# Structural insights into POT1-TPP1 interaction and POT1 C-terminal mutations in human cancer

Cong Chen[1,2,*], Peili Gu[3,*], Jian Wu[1,2], Xianyun Chen[1,2], Shuangshuang Niu[1,2], Hong Sun[1,2], Lijie Wu[1,2], Na Li[1,2], Junhui Peng[4], Shaohua Shi[1,2], Cuiying Fan[1,2], Min Huang[1,2], Catherine C.L. Wong[1,2], Qingguo Gong[4], Chandan Kumar-Sinha[5], Rongguang Zhang[1,2], Lajos Pusztai[6], Rekha Rai[3], Sandy Chang[3,7,8] & Ming Lei[1,2]

Mammalian shelterin proteins POT1 and TPP1 form a stable heterodimer that protects chromosome ends and regulates telomerase-mediated telomere extension. However, how POT1 interacts with TPP1 remains unknown. Here we present the crystal structure of the C-terminal portion of human POT1 (POT1C) complexed with the POT1-binding motif of TPP1. The structure shows that POT1C contains two domains, a third OB fold and a Holliday junction resolvase-like domain. Both domains are essential for binding to TPP1. Notably, unlike the heart-shaped structure of ciliated protozoan *Oxytricha nova* TEBPα–β complex, POT1–TPP1 adopts an elongated V-shaped conformation. In addition, we identify several missense mutations in human cancers that disrupt the POT1C–TPP1 interaction, resulting in POT1 instability. POT1C mutants that bind TPP1 localize to telomeres but fail to repress a DNA damage response and inappropriate repair by A-NHEJ. Our results reveal that POT1 C terminus is essential to prevent initiation of genome instability permissive for tumorigenesis.

[1] National Center for Protein Science Shanghai, State Key Laboratory of Molecular Biology, Institute of Biochemistry and Cell Biology, Shanghai Institutes for Biological Sciences, Chinese Academy of Sciences, 333 Haike Road, Shanghai 201210, China. [2] Shanghai Research Center, Chinese Academy of Sciences, Shanghai 200031, China. [3] Department of Laboratory Medicine, Yale School of Medicine, New Haven, Connecticut 05620, USA. [4] National Laboratory for Physical Science at the Microscale and School of Life Sciences, University of Science and Technology of China, Hefei 230026, China. [5] Department of Pathology, University of Michigan School of Medicine, Ann Arbor, Michigan 48109, USA. [6] Department of Medicine, Yale School of Medicine, New Haven, Connecticut 05620, USA. [7] Department of Pathology, Yale School of Medicine, New Haven, Connecticut 05620, USA. [8] Department of Molecular Biophysics and Biochemistry, Yale School of Medicine, New Haven, Connecticut 05620, USA. * These authors contributed equally to this work. Correspondence and requests for materials should be addressed to S.C. (email: s.chang@yale.edu) or to M.L. (email: leim@sibcb.ac.cn).

Telomeres are specialized protein–DNA complexes that cap the ends of linear eukaryotic chromosomes. Telomeric DNAs are composed of noncoding tandem repeats of a short G-rich sequence oriented 5′ to 3′ towards the chromosome terminus[1]. The protrusion of the G-rich strand forms a 3′ single-stranded (ss) overhang, which is conserved from simple eukaryotes to vertebrates[2–4]. Telomeres prevent the chromosome ends from activating DNA damage responses (DDRs)[5]. Defects in this protection lead to the initiation of DNA damage checkpoint cascades and DNA repair activities that cause chromosomal end-to-end fusions[6]. In most eukaryotes, telomeres provide a solution to the end-replication problem with telomerase, a specialized reverse transcriptase, adding telomeric repeats to the chromosome ends to ensure complete genome replication[1,7]. Dysregulation of telomere end protection by interfering with telomerase activities has been shown to promote the genomic instability associated with human diseases[8–14].

Human telomeres are protected by a specialized six-protein complex, shelterin[5]. In shelterin, TRF1 and TRF2 directly bind the duplex region of telomeres, and RAP1 is associated to telomere by interacting with TRF2. POT1, in a complex with TPP1, binds the 3′ ss overhang in a sequence-specific manner. TIN2 simultaneously interacts with TRF1, TRF2 and TPP1, thus serving as an interaction hub of the shelterin complex. POT1 and TPP1 function together by forming a stable heterodimer that protects chromosome ends and regulates telomerase activity[15–18]. There are two POT1 paralogs in mouse, mPOT1a and mPOT1b. mPOT1a functions primarily to repress an ataxia telangiectasia and RAD3 related (ATR)-dependent DDR at telomeres, while mPOT1b is required to repress nucleolytic processing of the 5′ telomeric C-strand. The single human POT1 possesses both of these functions[19–24]. POT1 was initially identified through its limited sequence similarity to the α-subunit of the ciliated protozoan *Oxytricha nova* TEBPα–β complex, the first characterized telomeric ssDNA-binding protein complex[4]. Similar to TEBPα, the N-terminal half of POT1 contains two oligosaccharide/oligonucleotide (OB) folds that specifically recognize telomeric ssDNA sequence[24,25]. TPP1 also contains an N-terminal OB fold that structurally highly resembles to the OB fold of TEBPβ[15]. The crystal structures of the two N-terminal OB folds of POT1 bound with telomeric ssDNA and the OB fold of TPP1 suggested that POT1 and TPP1 are the homologues of TEBPα and β subunits, respectively[4,15,24–27]. Both TEBPα and POT1 interact with their partner proteins—TEBPβ and TPP1—through their C-terminal protein–protein interaction domains. TEBPα employs a third OB fold at the C terminus to interact with an extended loop of TEBPβ[27]. However, the C-terminal TPP1-binding region of POT1 is twice as large as the third OB fold of TEBPα with no detectable sequence similarity[28]. Therefore, it is unclear whether the similarity between POT1 and TEBPα could be extended to their C-terminal regions.

TPP1 is the most versatile factor in the shelterin complex, playing several diverse roles in telomere maintenance and regulation. First, TPP1 interacts with both TIN2 and POT1, forming an intimate connection within the shelterin complex between the duplex and ss telomeric DNAs[17,29,30]. Second, the POT1–TPP1 heterodimer tightly associates with telomeric ss overhang and protects the chromosome ends from hazardous activities[17,29]. Finally, TPP1 actively recruits telomerase to telomeres and, together with POT1, functions as a processivity factor for telomerase during telomere extension[15]. A seven amino-acid cluster on the surface of TPP1 OB fold, the TEL patch, directly interacts with the TEN domain of telomerase reverse transcriptase (TERT) to recruit telomerase to telomeres[19,31,32].

Recent whole-genome and -exome sequencing have identified recurrent human POT1 mutations in chronic lymphocytic leukaemia[33,34], familial melanoma (FM)[35,36], glioma[37], cardiac angiosarcoma[38] and mantle cell lymphoma[39], making it the most commonly mutated shelterin component in cancer. Interestingly, the majority of these POT1 mutations cluster preferentially in the two N-terminal OB folds. Since POT1's ability to repress activation of DNA damage signalling and repair requires both OB folds[17,18,22,40], it is thought that OB fold mutations promote genome instability permissive for tumorigenesis[41]. However, a Q623H mutation in the POT1 C terminus identified in FM raises the intriguing possibility that the POT1 C terminus might play a role in maintaining telomere stability distinct from that of the N-terminal two OB folds[36].

Here we use structural and biochemical analyses to show that the C-terminal half of POT1 contains two domains, a third OB fold separated by a Holliday junction resolvase-like domain. Both domains are essential for binding to TPP1. Instead of a heart-shaped TEBPα–β-like structure, the POT1–TPP1 complex adopts an elongated V-shaped conformation. In addition, we identify C-terminal-specific POT1 mutations in human triple-negative breast cancers (TNBC). Several of these mutations form misfolded proteins incapable of interacting with TPP1, preventing localization to telomeres, while others are unable to repress end-to-end chromosome fusions through A-NHEJ-mediated repair. Our results reveal that in addition to the N-terminal OB folds, the C terminus of POT1 is also required to maintain genome stability to prevent cancer initiation.

## Results

**The POT1C–TPP1PBM complex structure.** Consistent with previous studies, yeast two-hybrid analysis showed that POT1 N-terminal two OB folds are not involved in the interaction with TPP1 (Supplementary Fig. 1a)[17]. Further analysis of the interaction between TPP1 and the C-terminal portion of POT1 (POT1C, residues 320–634) revealed that a short and highly conserved fragment of TPP1 consisting only of residues 266–320 was necessary and sufficient for binding with POT1C (Fig. 1a,b and Supplementary Fig. 1b). Hereafter, we will refer to TPP1$_{266–320}$ as TPP1$_{PBM}$ (POT1-binding motif) (Fig. 1a). We crystallized the POT1C–TPP1$_{PBM}$ complex and determined its structure by single-wavelength anomalous dispersion (SAD) at a resolution of 2.1 Å (Fig. 1c) (Supplementary Table 1 and Supplementary Fig. 2). The TPP1$_{PBM}$ polypeptide forms an extended structure that wraps around the surface of POT1C (Fig. 1c). The formation of the binary complex causes the burial of ∼2,100 Å$^2$ of surface area at the interface.

Although primary sequence analysis failed to identify any known protein motif in POT1C, the crystal structure reveals a typical OB-fold architecture comprising a highly curved five-stranded β-barrel at one end of POT1C (Fig. 1c). Because POT1 contains two additional ssDNA-binding OB folds at its N terminus, we will refer to the C-terminal OB fold as POT1$_{OB3}$ (Fig. 1a,c)[24]. The crystal structure also reveals a second domain at the other end of POT1C (residues 393–538), which adopts a compact, globular fold featuring a central curved seven-stranded β-sheet surrounded by four α helices (Fig. 1c). An unbiased search for structurally homologous proteins using Dali[42] revealed unequivocal structural resemblance of POT1$_{393–538}$ with archaeal Holliday junction resolvase Hjc (Fig. 1c and Supplementary Fig. 3a)[43]. The two structures can be superimposed onto each other with a root-mean-square deviation (r.m.s.d.) of ∼3.3 Å in the positions of over 80 Cα atoms of equivalent residues (Supplementary Fig. 3a). Hereafter, we will refer to POT1$_{393–538}$ as POT1$_{HJRL}$ (POT1 Holliday junction resolvase-like domain) (Fig. 1a). Despite the overall structural similarity of the central β-sheet, the peripheral regions

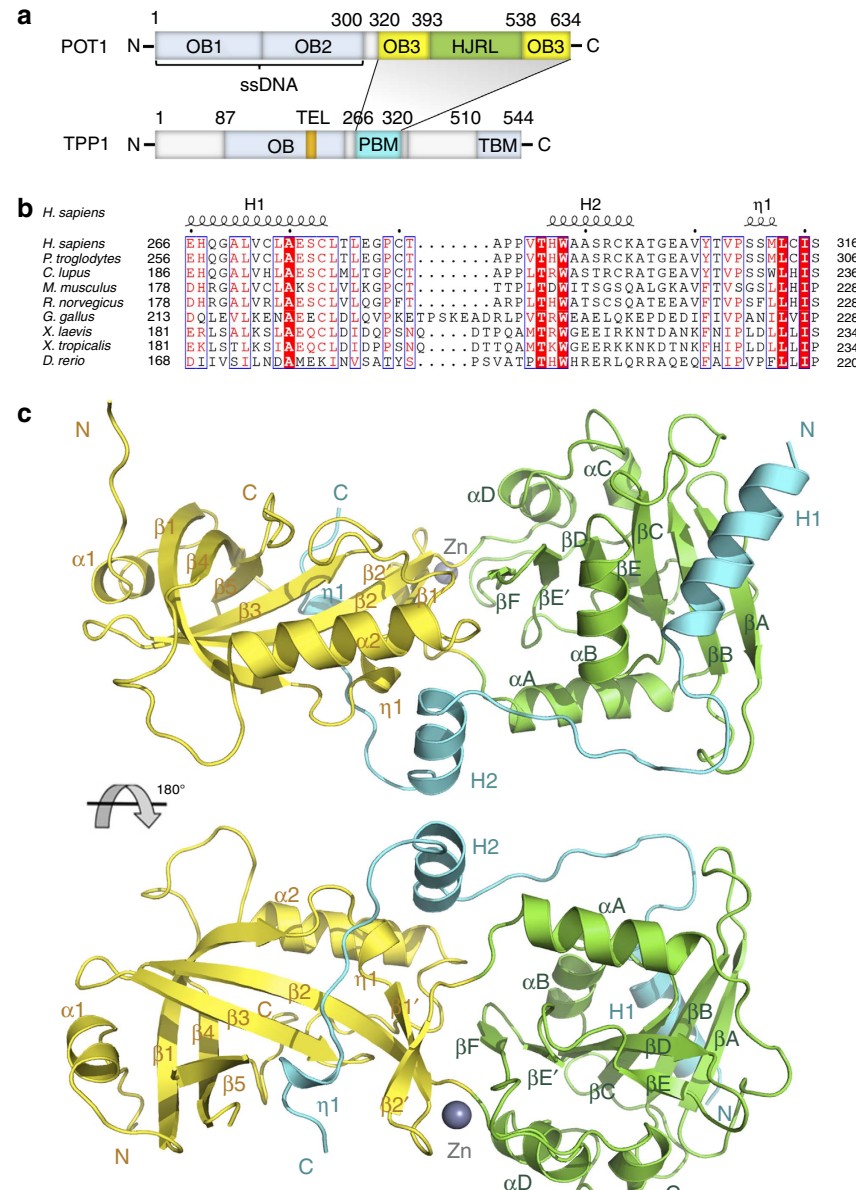

**Figure 1 | The POT1C–TPP1$_{PBM}$ complex structure.** (**a**) Domain organization of the POT1–TPP1 complex. OB1 and OB2 of POT1, the OB fold and the TBM (TIN2-binding motif) of TPP1 are coloured in light blue. The C-terminal OB3 and the embedded HJRL domain are coloured in yellow and green, respectively. The PBM of TPP1 is in cyan. The TEL patch in TPP1$_{OB}$ is coloured in orange. The shaded area indicates the interaction between POT1 and TPP1. (**b**) Structural-based sequence alignment of the PBM of human TPP1 and its homologues. The secondary structures (H, α-helix; η, 3$_{10}$-helix) of human TPP1 are labelled on the top. *Homo sapiens*, NP_001075955.1; *Pan troglodytes*, XP_003315229.1; *Canis lupus*, XP_853906.2; *Mus musculus*, NP_001012656.1; *Rattus norvegicus*, NP_001032270.1; *Gallus gallus*, XP_004944139.1; *Xenopus laevis*, NP_001089068.1; *Xenopus tropicalis*, NP_001120423.1; *Danio rerio*, NP_001124265.1. (**c**) Ribbon diagram of two orthogonal views of the POT1C–TPP1$_{PBM}$ complex. POT1$_{OB3}$ is coloured in yellow, POT1$_{HJRL}$ in green and TPP1$_{PBM}$ in cyan.

of POT1$_{HJRL}$ and Hjc are markedly divergent. Most notably, two short β strands in Hjc that is crucial for the interaction with Holliday junction are replaced by a long α-helix in POT1$_{HJRL}$ that is not compatible with Holliday junction binding (Supplementary Fig. 3a). Thus, these structure analyses suggested that POT1 could not bind Holliday junctions. Electrophoretic mobility shift assays (EMSA) confirmed that POT1C–TPP1$_{PBM}$ indeed did not show any detectable interaction with Holliday junctions (Supplementary Fig. 3b).

The OB3-HJRL packing in POT1 involves extensive hydrophobic contacts, and buried a total surface area of 592 Å$^2$ (Supplementary Fig. 4). Notably, there is a zinc ion between the

OB fold and the HJRL domain of POT1, which is coordinated by four cysteine residues contributed by both POT1$_{OB3}$ and POT1$_{HJRL}$ (Fig. 1c and Supplementary Fig. 4). This zinc ion helps stabilize the relative orientation between POT1$_{OB3}$ and POT1$_{HJRL}$. The residues that form the interface including those that chelate the zinc ion are all highly conserved across vertebrate species (Supplementary Fig. 5), indicating that the arrangement of the two domains might be important for POT1 functions in all vertebrates.

**Structural conservation between OB3 folds of POT1 and TEBPα.** Functional and structural studies have established that

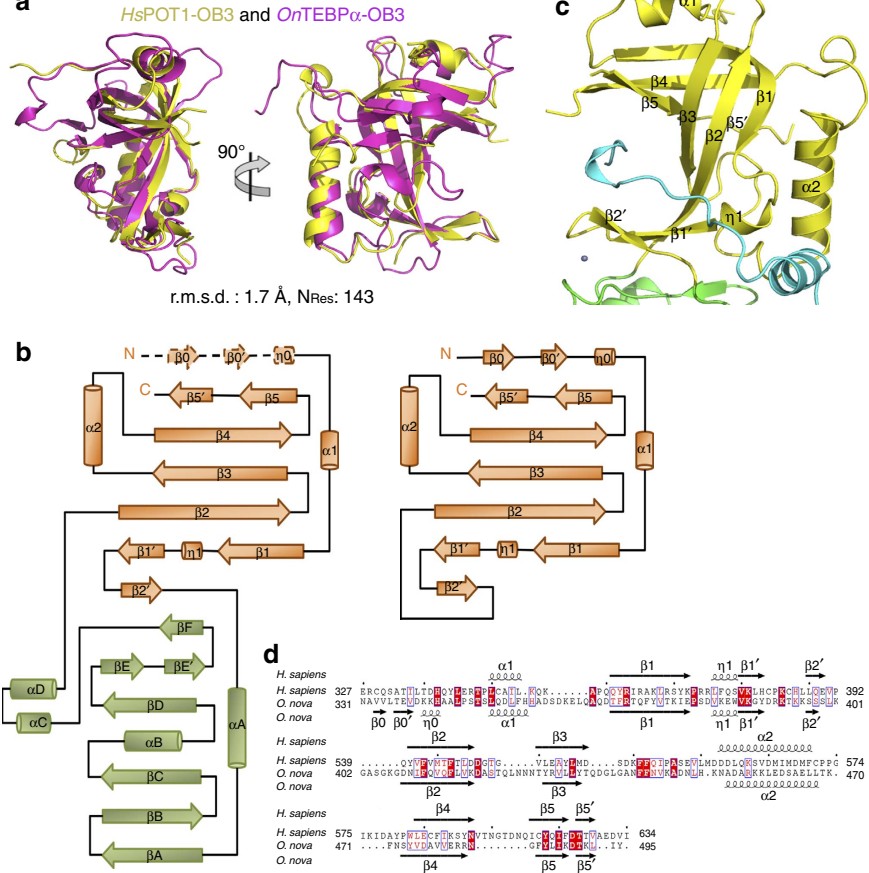

**Figure 2 | Structural conservation of POT1$_{OB3}$ and TEBPα$_{OB3}$.** (**a**) Superposition of human POT1$_{OB3}$ and *O. nova* TEBPα$_{OB3}$. POT1$_{OB3}$ is coloured in yellow and TEBPα$_{OB3}$ in magenta. The r.m.s.d. value and the number of residues used for superposition are listed. (**b**) Comparison of the topology diagram of C terminus of POT1 and TEBPα. (**c**) Similar to *O. nova* TEBPα$_{OB3}$, POT1$_{OB3}$ utilizes the canonical ssDNA-binding groove to bind a short 3$_{10}$-helix η1 of TPP1$_{PBM}$. (**d**) Sequence alignment of POT1$_{OB3}$ and TEBPα$_{OB3}$. Secondary structures (α, α-helix; β, β-strand; η, 3$_{10}$-helix) of human POT1$_{OB3}$ and *O. nova* TEBPα$_{OB3}$ are labelled on the top and bottom of the sequences, respectively.

POT1–TPP1 is the human homologue of the *O. nova* TEBPα–β complex; POT1 N-terminal OB1 and OB2, and TPP1$_{OB}$ closely resemble their counterparts in the TEBPα–β complex[4,15,24]. Consistently, Dali search revealed that the structure of POT1$_{OB3}$ is most similar to that of the C-terminal OB (OB3) fold of TEBPα[27,42]. The two OB folds can be superimposed with an r.m.s.d. of 1.7 Å in the positions of 143 equivalent Cα atoms (Fig. 2a). In addition to the overall structural similarity, the OB3 folds of POT1 and TEBPα share several unique features. First, there is a 3$_{10}$-helix insertion in the middle of strand β1, which introduces a sharp kink in β1 and makes room for the interacting partners TPP1 and TEBPβ (Fig. 2b,c). Second, in both POT1 and TEBPα, short strand β1′ after the kink is followed by another short strand β2′ on the opposite side of β2; the three strands together form a protruding arm of the β barrel (Fig. 2b,c). Third, in addition to OB3 folds themselves, both POT1$_{OB3}$ and TEBPα$_{OB3}$ utilize their canonical concaved side of the β barrels to bind a short helical structure of their interacting partners TPP1 and TEBPβ (Fig. 2c). Collectively, these structural similarities further support the notion that POT1 is the human homologue of *O. nova* TEBPα.

Despite the high degree of structural conservation, the sequences of OB3 folds of POT1 and TEBPα are quite divergent and share only 14% identity (Fig. 2d). Significant sequence and structural variation is particularly evident in the connecting loop regions. Most notably, the entire POT1$_{HJRL}$ domain

(146 residues) is embedded in the middle of POT1$_{OB3}$ between strands β2′ and β2 (Fig. 2b). In contrast, strands β2′ and β2 of TEBPα OB3 are connected by a nine-residue turn (Fig. 2b). These marked variances in the primary sequence explain the failure of identifying POT1$_{OB3}$ and detecting the similarity between OB3 folds of POT1 and TEBPα via bioinformatics approaches.

**XL-MS analysis of the POT1–TPP1 complex.** The crystal structure of the *O. nova* TEBPα–β–ssDNA complex reveals that the α and β subunits of TEBP associate with each other to form a heart-shaped structure that sandwiches the telomeric ssDNA in the middle (Fig. 3a)[27]. Although every OB fold in TEBPα–β is conserved in the POT1–TPP1 complex, it is still not clear whether the overall architecture of the POT1–TPP1 complex resembles that of TEBPα–β. To address this issue, we first superposed the crystal structures of TPP1$_{OB}$, POT1$_{OB3-HJRL}$–TPP1$_{PBM}$ and POT1$_{OB1-OB2}$–ssDNA onto the structure of the TEBPα–β–ssDNA complex (Fig. 3b). Notably, the structural overlay revealed that the distance between the end of TPP1$_{OB}$ (His241) and the beginning of TPP1$_{PBM}$ (Gly264) is ~80 Å if both POT1 and TPP1 occupy similar locations in the POT1–TPP1 complex as in TEBPα–β (Fig. 3b). Such a distance is too large for the 22-residue loop of TPP1 (242–263) that has to travel through the surface of POT1$_{OB3-HJRL}$ to connect TPP1$_{OB}$ and TPP1$_{PBM}$. Thus, this

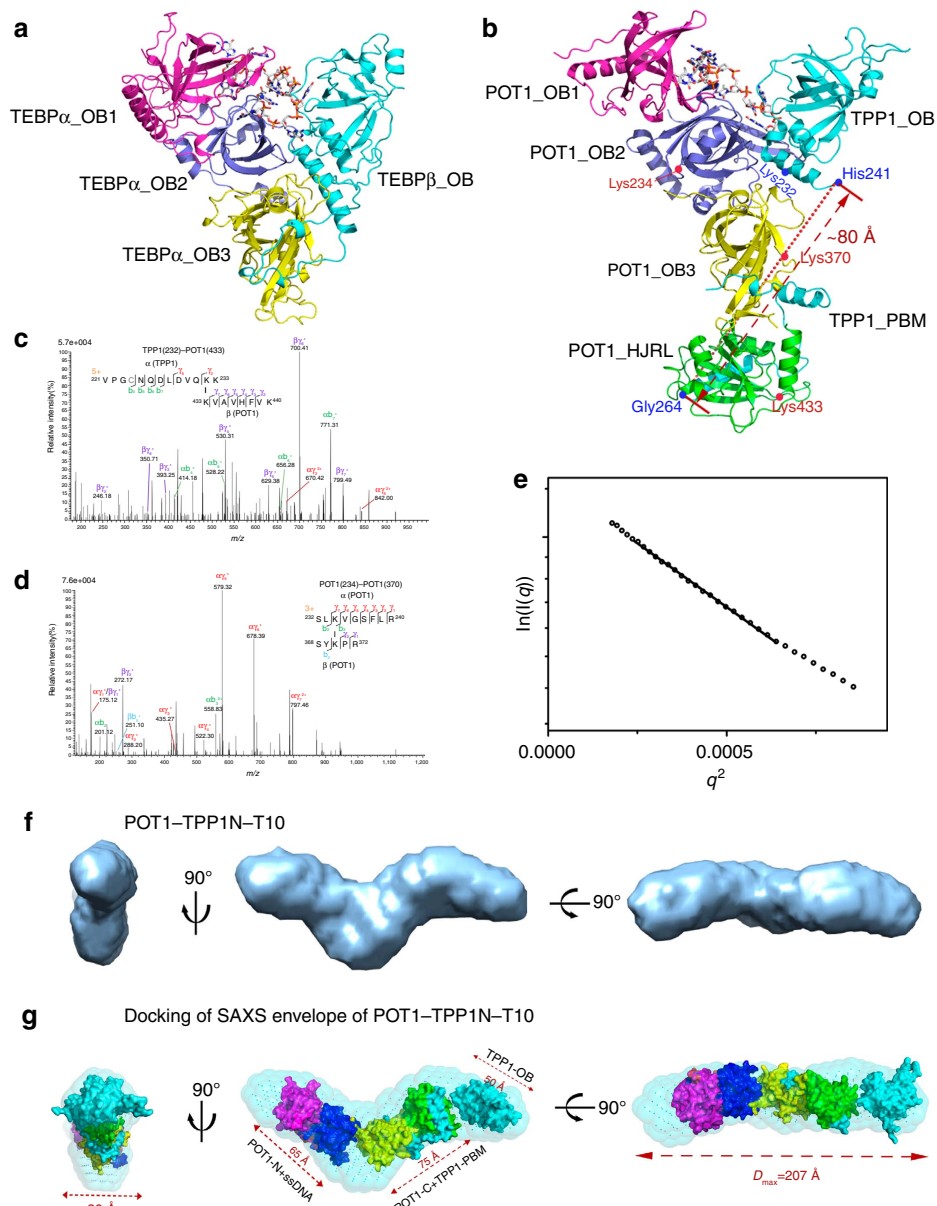

**Figure 3 | XL-MS and SAXS analyses indicate that the overall architecture of the POT1–TPP1–ssDNA complex is different from that of the TEBPα-β-ssDNA complex.** (**a**) Ribbon diagram of the heart-shaped TEBPα-β-ssDNA complex. (**b**) Superposition of the crystal structure of human TPP1$_{OB}$, POT1$_{OB3-HJRL}$–TPP1$_{PBM}$ and POT1$_{OB1-OB2}$–ssDNA onto the structure of the *O. nova* TEBPα-β-ssDNA complex. The dotted line indicates the distance between the end of TPP1$_{OB}$ (His241) and the beginning of TPP1$_{PBM}$ (Gly264). Crosslinked residues of POT1 and TPP1 are denoted by red and blue dots, respectively. (**c**) Annotated high-scoring spectrum of the crosslinked POT1–TPP1N complex unambiguously identified the crosslinked peptide sequences as KVAVHFVK of POT1 and VPGCNQDLVQKK of TPP1, respectively, demonstrating an intermolecular linkage between POT1$^{Lys433}$ and TPP1$^{Lys232}$. (**d**) Annotated high-scoring spectrum of the crosslinked POT1–TPP1N complex unambiguously identified the crosslinked peptide sequences as SLKVGSFLR and SYKPR of POT1, respectively, demonstrating an intramolecular linkage between residues Lys234 and Lys370 of POT1. (**e**) Guinier plot of the POT1–TPP1N–T10 complex indicating that the complex is monodisperse and homogeneous in solution. (**f**) Three views of the V-shaped envelop of the POT1–TPP1N–T10 complex. The envelope is coloured in light blue. (**g**) Docking of the crystal structures of human TPP1$_{OB}$, POT1$_{OB3-HJRL}$–TPP1$_{PBM}$ and POT1$_{OB1-OB2}$–ssDNA into the envelope of POT1–TPP1N–T10 complex. POT1 OB1, OB2, OB3, and HJRL and TPP1 are coloured in magenta, blue, yellow, green and cyan, respectively. Envelope of the POT1–TPP1N–T10 complex is coloured in light blue.

analysis suggests that the overall architecture of the POT1–TPP1 complex is very likely different from that of TEBPα-β.

We next employed crosslinking mass spectrometry (XL-MS) to examine the spatial relationship between POT1 and TPP1 in the complex. The N-terminal 86 residues of human TPP1 are not conserved in other species[15], thus hereafter TPP1Δ(1–86) will be referred to as TPP1, unless stated otherwise. Our previous study showed that the N-terminal half of TPP1 (TPP1N, residues 87–334) is sufficient for the interaction with POT1 and telomeric ssDNA[15]. Efficient ssDNA binding of POT1–TPP1N required a ssDNA of 10 bases or longer that included the core telomeric sequence 5′-TTAGGGTTAG-3′ (T10)[15,24]. We purified the POT1–TPP1N–T10 ternary complex and subjected it to chemical crosslinking using disuccinimidyl suberate. The cross-linked product was digested by trypsin and then analysed by liquid chromatography-mass spectrometry (Fig. 3c,d). Analysis of

the MS data unambiguously identified an intermolecular link between POT1[Lys433] and TPP1[Lys232] (Fig. 3c). Although POT1[Lys433] and TPP1[Lys232] might not be directly involved in mediating the POT1–TPP1 interaction, they are close to each other surrounding the interface. Notably, POT1[Lys433] is located at the distal end of POT1[HJRL] away from POT1[OB3], whereas TPP1[Lys232] is on the C-terminal α3-helix of TPP1[OB] (Fig. 3b). Therefore, TPP1[OB] must be in close vicinity to the HJRL domain but not the OB3 fold of POT1. This is consistent with our structural superposition analysis that TPP1[OB] does not occupy the same location as in the TEBPα–β complex (Fig. 3b). In addition, we also detected an intramolecular link between POT1[Lys234] and POT1[Lys370] (Fig. 3d). These two lysine residues are, respectively, located in the second and the third OB folds of POT1 (Fig. 3b), suggesting that POT1[OB2] and POT1[OB3] are spatially close to each other.

**SAXS structural analysis of the POT1–TPP1 complex.** Next, we analysed the three-dimensional (3D) structure of the POT1–TPP1N–T10 complex using small-angle X-ray scattering (SAXS) (Supplementary Fig. 6). The Guinier region of the scattering curve is linear (Fig. 3e), indicating that the POT1–TPP1N–T10 complex is monodisperse and homogeneous in solution. The SAXS data were used to calculate the maximum particle dimension ($D_{max}$) and radius of gyration ($R_g$), which showed highly similar values at all evaluated concentrations for the POT1–TPP1N–T10 complex (Supplementary Table 2).

The experimental SAXS data were next used for the reconstruction of 20 individual *ab initio* molecular envelopes using dummy bead modelling in the programme DAMMIF[44]. The most representative model was picked as the one having the lowest normalized spatial discrepancy compared to the rest of the models (Fig. 3f). The derived envelope demonstrates that the POT1–TPP1N–T10 complex adopt an elongated V-like topology with one arm longer than the other (Fig. 3f). Based on the available crystal structure, the longer dimensions of POT1[OB1-OB2]–T10, POT1[OB3-HJRL]–TPP1[PBM] and TPP1[OB] are ∼ 65 Å, 75 Å and 50 Å, respectively (Fig. 3g)[15,24]. Notably, the sum of these three distances is consistent with the experimentally determined $D_{max}$ value and the longer dimension of the envelope of the POT1–TPP1N–T10 complex (Supplementary Table 2), strongly suggesting that the three structural modules are linearly connected in the complex. In keeping with this idea, the shorter dimensions of POT1[OB1-OB2]–T10, POT1[OB3-HJRL]–TPP1[PBM] and TPP1[OB] are all about 30 Å, which matches well with the narrow dimension of the envelope of the POT1–TPP1N–T10 complex (Fig. 3g)[15,24].

Based on the elongated shape of the POT1–TPP1N–T10 complex and the XL-MS results that POT1[OB3] and POT1[HJRL] are, respectively, close to POT1[OB2] and TPP1[OB] (Fig. 3c,d), we conclude that POT1[OB3-HJRL] must occupy the middle position within the V-shaped envelope, whereas POT1[OB1-OB2]–T10 and TPP1[OB] take up the two ends. We manually docked the crystal structures of POT1[OB1-OB2], POT1[OB3-HJRL]–TPP1[PBM] and TPP1[OB] into the envelope of the POT1–TPP1N–T10 complex (Fig. 3g). We fit TPP1[OB] and POT1[OB3-HJRL]–TPP1[PBM] into the longer arm and POT1[OB1-OB2]–T10 into the short arm of the V-shaped envelop, resulting in a model for the envelop of SAXS data (Fig. 3g). Clearly, this V-shaped human POT1–TPP1N–T10 complex is substantially different from the *O. nova* heart-shaped TEBPα–β–ssDNA structure (compare Fig. 3a,g).

To validate this model, we next analysed the 3D structure of the POT1–TPP1[PBM]–T10 complex using SAXS (Supplementary Fig. 7a–d). Similar to the POT1–TPP1N–T10 complex, the derived envelope of POT1–TPP1[PBM]–T10 also exhibited

a V-shaped volume with two roughly symmetric arms (Supplementary Fig. 7e). The dimensions of the envelope were consistent with the experimentally determined $R_g$ and $D_{max}$ values (Supplementary Table 3). We docked the envelope of POT1–TPP1[PBM]–T10 into that of POT1–TPP1N–T10 to determine the position of TPP1[OB], the only difference between the two complexes. The docking resulted in a very good fit with a map correlation of 0.95; the arms of the two V-shaped envelopes aligned well with each other (Supplementary Fig. 7f). The superposed envelopes showed that POT1–TPP1N–T10 exhibited a larger volume than POT1–TPP1[PBM]–T10 with additional density of TPP1[OB] at the end of the longer arm of the POT1–TPP1N–T10 envelope (Supplementary Fig. 7f). Notably, this position of TPP1[OB] is consistent with the model of the POT1–TPP1N–T10 complex (Supplementary Fig. 7g), demonstrating that the model is correctly fitted into the envelope and the POT1–TPP1N–T10 complex indeed adopts a V-shaped conformation in solution.

**Interactions between POT1C and TPP1[PBM].** The structure of the POT1C–TPP1[PBM] complex reveals the molecular basis by which TPP1 recognizes POT1. TPP1[PBM] contains two α-helices (H1 and H2) and one 3$_{10}$-helix (η1) (Fig. 1b,c). Accordingly, it can be roughly divided into three binding modules, which all form extensive contacts with POT1C. The H1 helix of TPP1[PBM] consists of residues Glu266 to Cys278. The topography of the complementary POT1 molecular surface is rather hydrophobic in nature with a shallow groove formed by the curved β-sheet and helix αB of the HJRL domain (Fig. 4a,b). Four hydrophobic residues in helix H1 of TPP1[PBM], Leu271, Ala275, Leu279 and Leu281 make intimate interactions to the POT1 groove (Fig. 4a,b). The main chain carbonyl and amino groups of TPP1 Thr280 coordinate with the side chains of Arg432 and Glu461 of POT1, respectively (Fig. 4b). These interactions act as tethers to stabilize the relative positioning of the TPP1[PBM] H1 helix on POT1.

The H2 helix of TPP1 fits into a depression between POT1[OB3] and POT1[HJRL] opposite to the Zn ion (Fig. 4c,d). A conserved residue Trp293 at the beginning of helix H2 fits snugly into a hydrophobic pocket formed by helix α2 and 3$_{10}$-helix η1 of POT1[OB3] (Fig. 4c,d). This configuration is further stabilized by hydrogen-bonding interactions among TPP1[His292], TPP1[Trp293], POT1[Asp577] and POT1[Asp584] (Fig. 4c). In the middle of the H2 helix, the side chain of TPP1[Arg297] points into a deep pocket between POT1[OB3] and POT1[HJRL] and makes a direct hydrogen bond with the main chain carbonyl of POT1[Val391] (Fig. 4c,d). The loop between helices H1 and H2 of TPP1[PBM] does not contribute to the interactions with POT1. This is consistent with the observation that this region is variable in both sequence and length and adopts a dynamic conformation in the crystal (Fig. 1b and Supplementary Fig. 8).

3$_{10}$-Helix η1 and its periphery residues at both sides compose the third interaction module of TPP1[PBM], fitting into a hydrophobic groove formed by the concaved side of POT1[OB3] β-barrel (Fig. 4e,f). This groove is the canonical ssDNA-binding site of the OB folds[24,27,45,46]. TPP1 residues Val305–Ser316 travel in this groove in a direction that is roughly perpendicular to strands β2 and β3 of POT1[OB3] (Fig. 4e,f). Tyr306, Val308 and Leu313 of TPP1 contribute most to the hydrophobic interactions (Fig. 4f). This hydrophobic core is further complemented by hydrogen-bonding interactions at the periphery (Fig. 4f). Together, these hydrophobic and electrostatic interactions stabilize the C-terminal portion of TPP1[PBM] in the groove of POT1[OB3].

To corroborate our structural analysis, we examined whether mutating TPP1 residues at the POT1–TPP1 interface could weaken or disrupt POT1–TPP1 interaction. First, we mutated

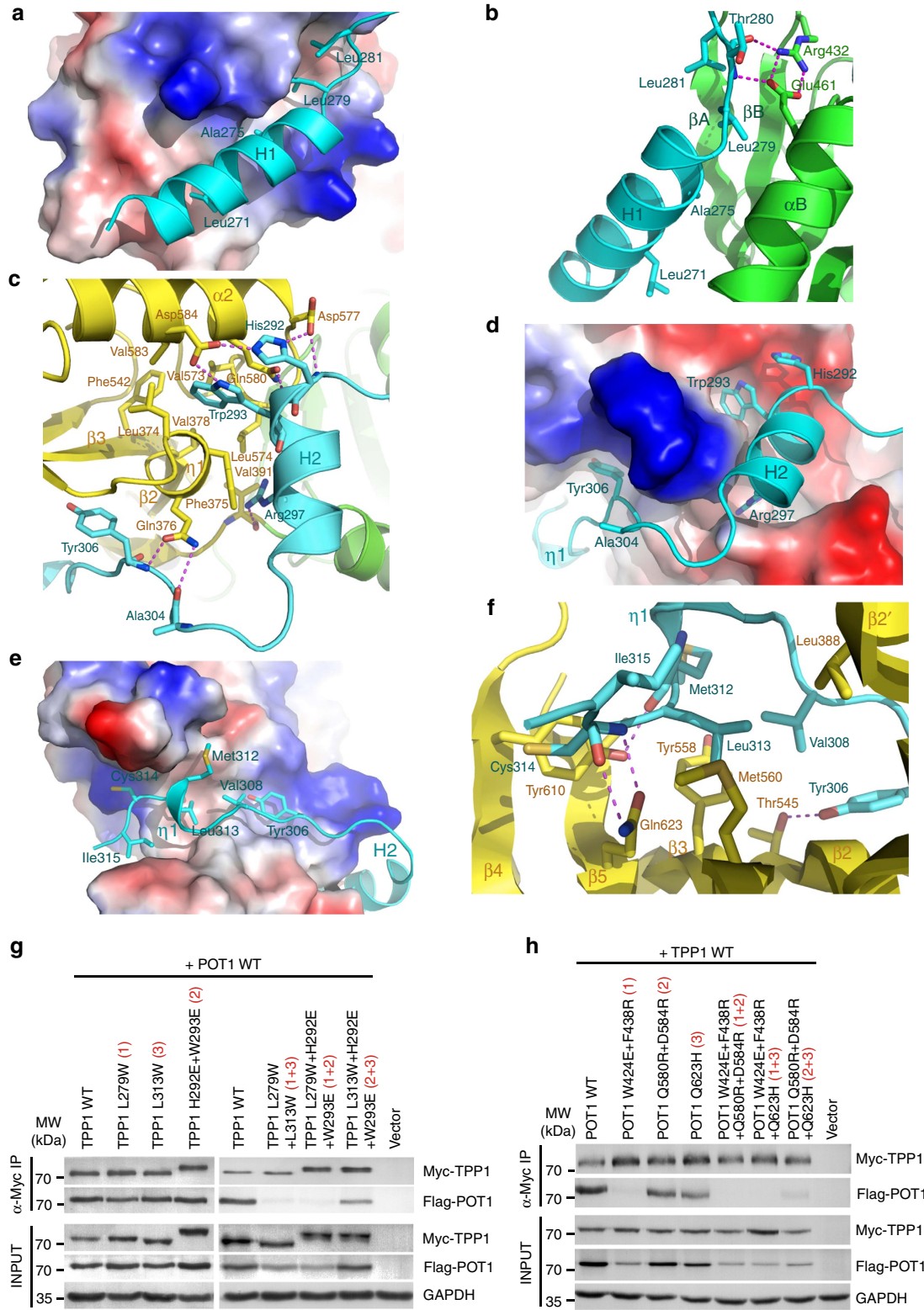

**Figure 4 | Structural and mutational analyses of the POT1C–TPP1$_{PBM}$ interaction.** (**a,d,e**) The electrostatic surface potential of the three TPP1$_{PBM}$-binding modules on POT1C (positive potential, blue; negative potential, red). TPP1$_{PBM}$ is in ribbon representation and coloured in cyan. (**b,c,f**) The intermolecular interactions at the three TPP1$_{PBM}$-binding modules interface. The colour scheme is the same as in Fig. 1c. Residues important for the interaction are shown as stick models. Salt bridges and hydrogen-bonding interaction are shown as magenta dashed lines. (**g**) Co-IP of POT1 with co-transfected WT and mutations of TPP1 that interfere with one or two POT1–TPP1-binding modules. The levels of each protein in the input and IP samples were analysed by immunoblotting with the indicated antibodies. 'Input' contains 5% of the input whole cell lysate used for IPs. (**h**) Co-IP of TPP1 with co-transfected WT and mutations of POT1 that interfere with one or two POT1–TPP1-binding modules.

key-interacting surface residues of TPP1 and examined their effects on binding to POT1 by co-immunoprecipitation (co-IP). Mutations that interfere with only a single binding module could not disrupt the interaction between POT1–TPP1 (Fig. 4g). However, TPP1 mutations that interfere with the first plus either the second or the third binding modules abolished the interaction between POT1 and TPP1 (Fig. 4g). In contrast, mutations that affect both the second and the third binding modules still maintained a weak interaction (Fig. 4g), suggesting that these two binding modules play a minor role in the POT1–TPP1 interaction. Next, we evaluated the effects of mutations of key-interacting residues of POT1 on its binding to TPP1. Notably, a double point mutation POT1[W424E/F438R] designed to disrupt only the first binding module completely abrogated the interaction between POT1 and TPP1, whereas mutations that interfere with the other two binding modules had only marginal effects (Fig. 4h). In addition, POT1[Q580R/D584R/Q623H] mutations that interfere with both the second and the third binding modules also abolished the POT1–TPP1 interaction (Fig. 4h). Parallel immunofluorescence telomere localization experiments were completely consistent with our co-IP data (Supplementary Fig. 9). Taken together, we conclude that although all three binding modules contribute to the POT1-TPP1 interaction, the first module is essential for the binding. Notably, IP analysis also showed that the steady-state protein levels of all TPP1-binding deficient mutants of POT1 were lower than that of wild type (WT) POT1 (Fig. 4g,h), suggesting that POT1 is unstable if it cannot associate with TPP1 in cells.

**Characterization of POT1C mutations in human cancers.** Given the importance of POT1C in its interaction with TPP1, we speculated that mutations within this domain might destabilize POT1 by preventing POT1–TPP1 complex formation. In addition to FM, we investigated the possibility that mutations in POT1C might also be present in other human cancers. We searched the catalogue of somatic mutations in the cancer and the Cancer Genome Atlas databases for POT1C mutations, and also performed whole-genome sequencing on 59 patients with TNBC (Jiang *et al.*, manuscript in preparation). We found four POT1C missense mutations (A364E, P371T, E572K and M587T) in advanced TNBC in these data sets at a frequency of ~6.7%. In contrast to the chronic lymphocytic leukaemia and FM POT1 mutations[33–36], all POT1C TNBC mutations disrupted phylogenetically conserved amino acids within OB3 (Fig. 5a and Supplementary Fig. 5). Interestingly, mutations within the POT1 N-terminal OB folds were not detected in these TNBCs.

To understand how TNBC POT1C mutations impacted on telomere function, we generated HA-tagged TNBC POT1C mutants as well as the C-terminal FM mutant POT1[Q623H] and compared their telomere protective functions with a N-terminal OB-fold mutant POT1[F62A] that has been previously shown to lack the ability to protect telomeres from activating a DDR[18]. Notably, all POT1 mutant proteins were unstable if TPP1 is not co-expressed in 293T cells (Fig. 5b). Co-expression of TPP1 increased the levels of WT POT1, the N-terminal OB1 mutant POT1[F62A] and the POT1C mutants POT1[E572K] and POT1[M587T] (Fig. 5b). In sharp contrast, POT1C mutants POT1[A364E] and POT1[P371T], which reside within the N-terminal portion of OB3, are expressed at very low levels even in the presence of TPP1 (Fig. 5b). Compared to WT POT1, POT1[Q623H] is also expressed at lower levels when co-expressed with TPP1, although not to the same extent as POT1[A364E] and POT1[P371T] (Fig. 5b). To determine whether this lower level of expression reflected increased protein turnover, we performed a cycloheximide (CHX) chase assay to block protein synthesis and monitored the decay of HA-tagged POT1 proteins by western analysis. Compared to WT POT1, the steady-state levels of POT1C mutants POT1[A364E] and POT1[P371T] were very low, even in the presence of TPP1 (Supplementary Fig. 10a). In addition, POT1[Q623H] also exhibited reduced half-life (Supplementary Fig. 10a).

Localization of POT1 to telomeres *in vivo* requires its interaction with TPP1, while POT1 N-terminal OB1 and OB2 folds are dispensable for telomere localization[17,18,20,23,29,30,40]. WT POT1, POT1[F62A] and POT1[E572K], POT1[M587T] and POT1[Q623H] all readily localized to telomeres (Fig. 5c,d). In contrast, in POT1[A364E] and POT1[P371T] expressing cells, telomere localization of POT1 mutant proteins was greatly reduced (Fig. 5c,d). This result is in agreement with our CHX data, demonstrating the inability to generate these mutants at high levels. To determine whether these POT1C mutants are defective in binding to ss telomeric DNA, we performed an *in vitro* telomere-binding assay using TTAGGG (Tel G) oligonucleotides in the presence of TPP1. WT POT1, POT1[E572K], POT1[M587T] and POT1[Q623H] all robustly interacted with Tel G (Fig. 5e). Even at reduced levels, mutants POT1[A364E] and POT1[P371T] still bound to Tel G (Fig. 5e). As a control, POT1[F62A] was completely unable to bind to Tel G (Fig. 5e)[18]. In addition, co-expression of WT POT1 with the dominant-negative mutant TPP1[ΔRD], which lacks the POT1 recruitment domain (residues 244–337)[23] and was unable to interact with POT1, resulted in reduced stability of WT POT1, but still showed some binding to Tel G (Fig. 5e). Taken together, these results reveal that the POT1C mutations did not impact on POT1's ability to interact with telomeric ssDNA.

We next asked whether POT1C mutants function as dominant negatives to induce a DDR at telomeres in U2OS cells, using the dysfunctional telomere-induced DNA damage foci (TIF) assay to monitor for the recruitment of DNA damage marker γ-H2AX to telomeres. As controls, more than five TIFs per cell were observed in ~40% of U2OS cells expressing either the POT1[F62A] mutant or TPP1[ΔRD] (Fig. 6a,b)[17]. In contrast, only ~5–10% of U2OS cells expressing mutants POT1[E572K], POT1[M587T] or POT1[Q623H] displayed ≥5 TIFs per cell (Fig. 6a,b). To further assess whether POT1C mutants are able to protect telomeres devoid of endogenous POT1 from engaging in a DDR, we reconstituted WT POT1, POT1[F62A] or the POT1C mutants, together with human TPP1, into *CAG-Cre[ER]; mPOT1a[F/F], mPOT1b[−/−]* MEFs. Addition of 4-hydroxytamoxifen (4-HT) initiated Cre-mediated deletion of the floxed *mPOT1a* allele, resulting in the elimination of endogenous mPOT1a/b proteins and robust TIF formation (Fig. 6c,d). We and others have previously shown that human POT1 can be readily reconstituted and its function characterized in mouse cells lacking both mPOT1a and mPOT1b in the presence of human TPP1 (refs 47–49). Consistent with previous data, expression of WT human POT1, but not POT1[F62A], efficiently repressed the occurrence of TIFs in mPOT1a/b double knockout MEFs (Fig. 6c,d). While reconstitution with POT1[E572K] and POT1[M587T] reduced the number of TIFs to levels observed in cells expressing WT POT1, expression of POT1[A364E], POT1[P371T] and POT1[Q623H] resulted in increased activation of a DDR at telomeres (Fig. 6c,d). Notably, expression of POT1[F62A] failed to suppress the TIF formation (Fig. 6c,d). Thus, our data confirm that the POT1 OB1 and OB2 are essential to repress the activation of a DDR at telomeres[18]. Nevertheless, the observation that the POT1C mutants induced even low level of TIFs is surprising, since the POT1 C terminus has never been previously shown to be required to repress a DDR at telomeres.

**POT1C mutations promote genome instability.** We next examined whether POT1C mutants promote the formation of end-to-end chromosome fusions, a hallmark of dysfunctional

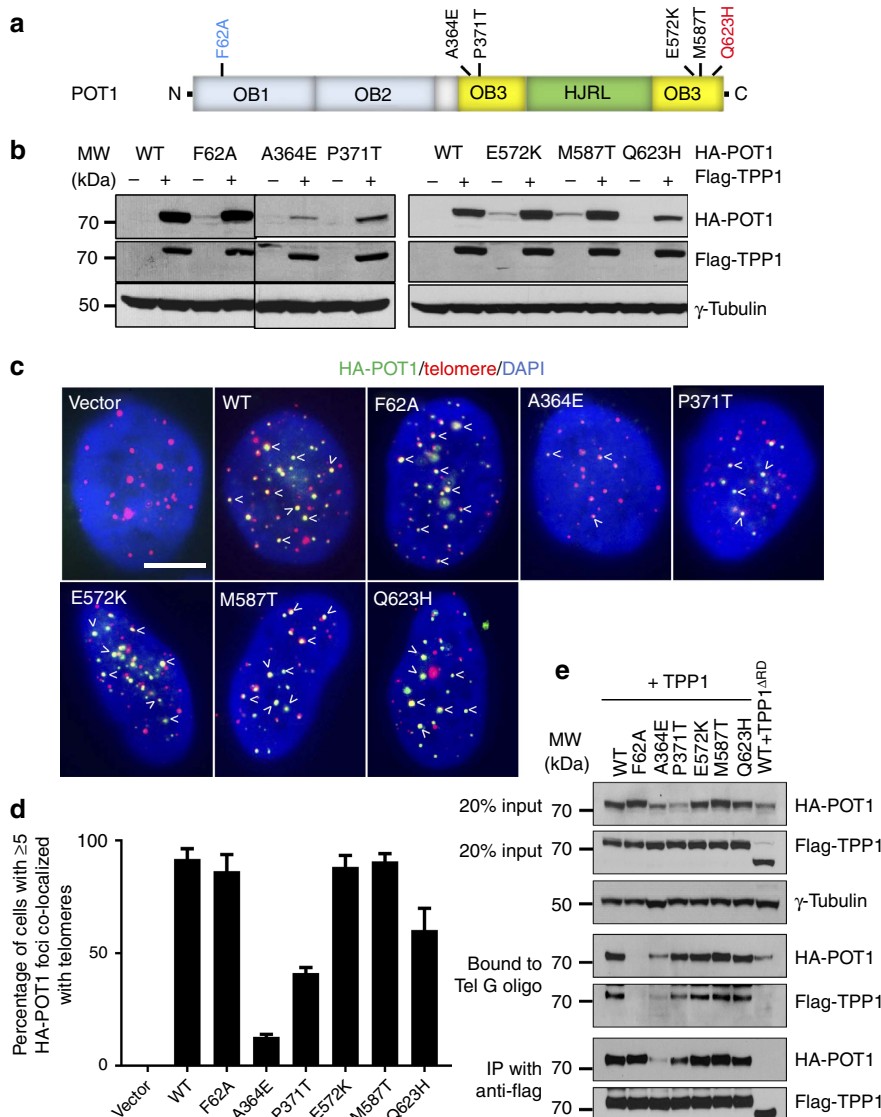

**Figure 5 | Characterization of POT1 C-terminal mutants. (a)** POT1C mutations identified in human cancers. The N-terminal OB1 mutation F62A is coloured blue, while POT1C mutations found in TNBC and FM are coloured black and red, respectively. **(b)** Co-expression of WT and mutant HA–POT1 with or without Flag-TPP1 in 293T cells. γ-tubulin served as a loading control. **(c)** Telomere co-localization of WT and mutant HA–POT1 in U2OS cells. Cells were immunostained with anti-HA (green), hybridized with Cy5-(CCCTAA)$_4$ probe (red) to detect telomeres and stained with DAPI (blue) for nuclei. Arrows point to HA–POT1 foci at telomeres. Scale bar, 5 μm. **(d)** Quantification of **c** to illustrate percentage of cells with more than five HA–POT1 foci co-localized with telomeres. Error bar (s.e.m.) was derived from three repeated experiments. A minimum of 250 nuclei were scored per experiment and three independent experiments were performed. **(e)** DNA binding and co-IP assays to assess the impact of HA–POT1 mutations on binding to ss Tel-G (TTAGGG)$_6$ oligonucleotides in the presence of Flag-TPP1. WT or mutant HA–POT1 proteins were co-expressed with either Flag-tagged WT TPP1 or TPP1$^{ΔRD}$ in 293T cells. Cell lysates were incubated with streptavidin beads bound by biotinylated ss Tel-G or anti-Flag conjugated agarose beads. Input represents 20% of lysate used for DNA binding or IP.

telomeres. End-to-end chromosome fusions were observed at approximately equal frequencies (∼1.8–2.5 fusions per 100 chromosomes analysed) in IMR90 cells expressing either the POT1$^{F62A}$ mutant or the three highly expressed POT1C mutants (Fig. 7a,b). Notably, this level of end-to-end fusion is higher than what was observed in telomerase knockout MEFs[50]. Interestingly, while fusion sites containing telomeric signals were always detected in cells expressing the POT1$^{F62A}$ mutant, they were never observed at fusion sites in cells expressing POT1C mutations (Fig. 7a,b). Since telomeres devoid of POT1 are repaired via the PARP1-mediated alternative-non-homologous end-joining (A-NHEJ) pathway[51], we asked whether the fusions observed in POT1C mutants are generated via A-NHEJ. Treatment with the PARP1 inhibitor PJ34

resulted in ∼50% reduction of chromosomal end-to-end fusions in IMR90 cells expressing either POT1$^{F62A}$ or the POT1C mutations, revealing that chromosome fusions stemming from expression of either POT1N or POT1C mutations are due to the activation of A-NHEJ-mediated DNA repair (Fig. 7c). Finally, to assess the role of POT1 mutations in protecting chromosome ends devoid of endogenous mPOT1a/b, we reconstituted 4-HT-treated *CAG-Cre$^{ER}$; mPOT1a$^{F/F}$, mPOT1b$^{-/-}$* MEFs with WT POT1, POT1N and POT1C mutants. While both POT1$^{E572K}$ and POT1$^{M587T}$ reduced the number of chromosomal aberrations to levels observed in cells expressing WT POT1, the POT1$^{A364E}$, POT1$^{P371T}$ and POT1$^{Q623H}$ mutants cannot repress aberrant A-NHEJ-mediated repair, resulting in cytogenetic aberrations

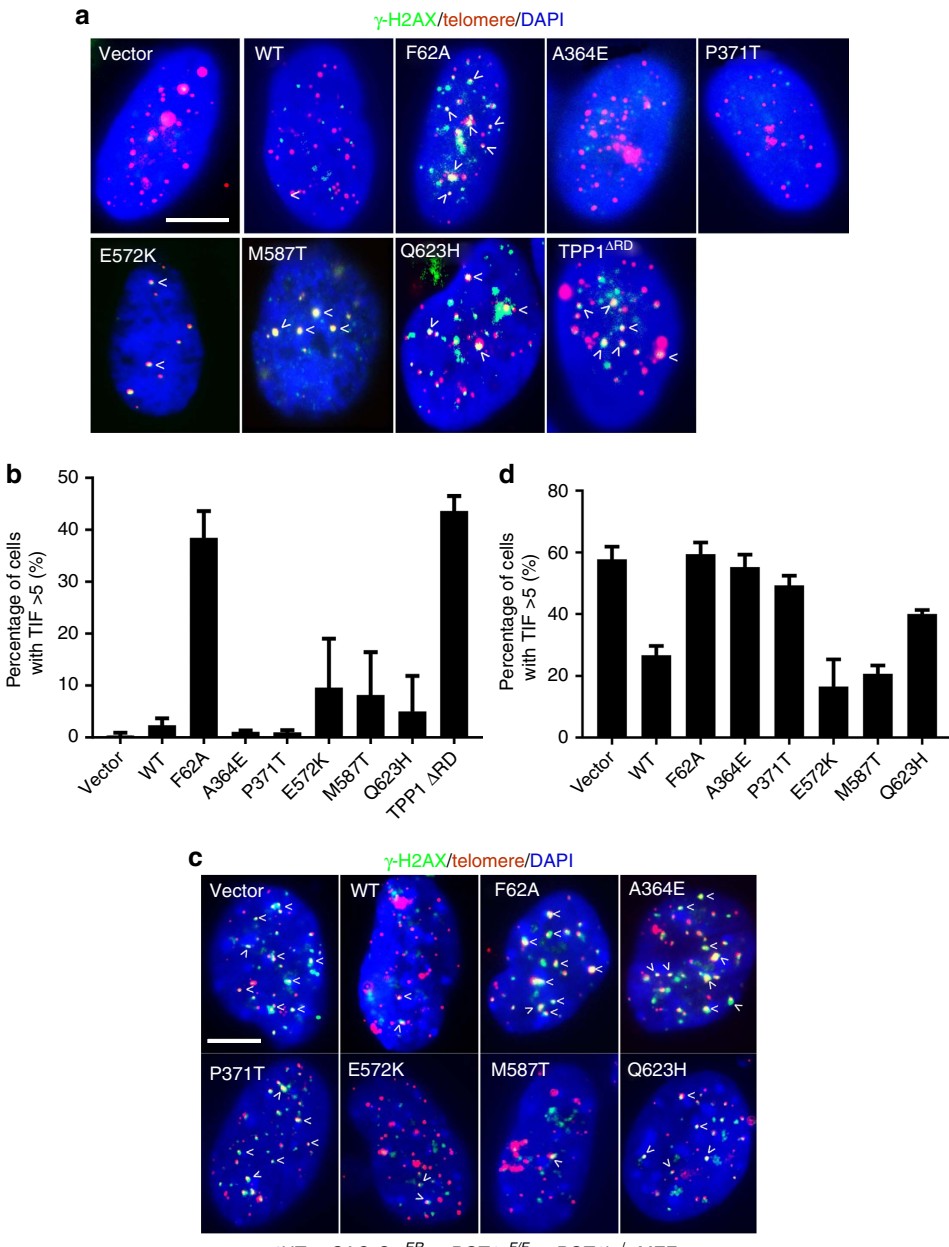

**Figure 6 | Some POT1C OB3 mutants cannot protect telomeres from engaging a DDR.** (**a**) Co-localization of γ-H2AX with telomeres in U2OS cells expressing WT or mutant HA–POT1. Cells were immunostained with anti-HA antibody to detect HA–POT1 (green), hybridized with Cy3-(CCCTAA)$_4$ probe (red) to detect telomeres and DAPI (blue) for nuclei. Arrows point to a few TIFs. Scale bar, 5 μm. (**b**) Quantification of frequency of γ-H2AX-positive TIFs in **a**. Cells with ≥5 TIFs were scored as positive. TPP1$^{\Delta RD}$ treatment to elicit TIF formation was used as a positive control. Error bar (s.e.m.) was derived from three repeated experiments. A minimum of 100 nuclei were scored per experiment and three independent experiments were performed. (**c**) Co-localization of γ-H2AX with telomeres in CAG-Cre$^{ER}$; mPOT1a$^{F/F}$, mPOT1b$^{-/-}$ MEFs reconstituted with the indicated DNA constructs and then treated with 4-HT. Cells were immunostained with anti-γ-H2AX antibody (green), hybridized with Cy3-(CCCTAA)$_4$ probe to detect telomeres (red) and DAPI (blue) for nuclei. Arrows point to a few TIFs. Scale bar, 5 μm. (**d**) Quantification of percentage of γ-H2AX-positive TIFs in **c**. Cells with ≥5 TIFs were scored as positive. TPP1$^{\Delta RD}$ treatment to elicit TIF formation was used as a positive control. Error bar (s.e.m.) was derived from three experiments with a minimum of 30 metaphases were scored per experiments.

approaching those observed in cells expressing POT1$^{F62A}$ (Fig. 7d and Supplementary Fig. 11a). In parallel, we also generated analogous mutations into mPOT1a and reconstituted WT or mPOT1a mutants into CAG-Cre$^{ER}$; mPOT1a$^{F/F}$, mPOT1b$^{-/-}$ MEFs and removed endogenous mPOT1a with 4-HT. Like their human counterparts, expression of both mutants mPOT1a$^{A370E}$ and mPOT1a$^{P377T}$ was reduced even in the presence of mTPP1 and these mutants cannot repress A-NHEJ-mediated repair

(Supplementary Figs 10b and 11b,c). In contrast, mPOT1a$^{Q629H}$ appears to be more stable than POT1$^{Q623H}$ and is thus able to rescue cytogenetic aberrations to WT levels (Supplementary Figs 10b and 11b–c).

**Structural implication of POT1C mutations in human cancers.** Our structural studies provide clues of possible structural impacts

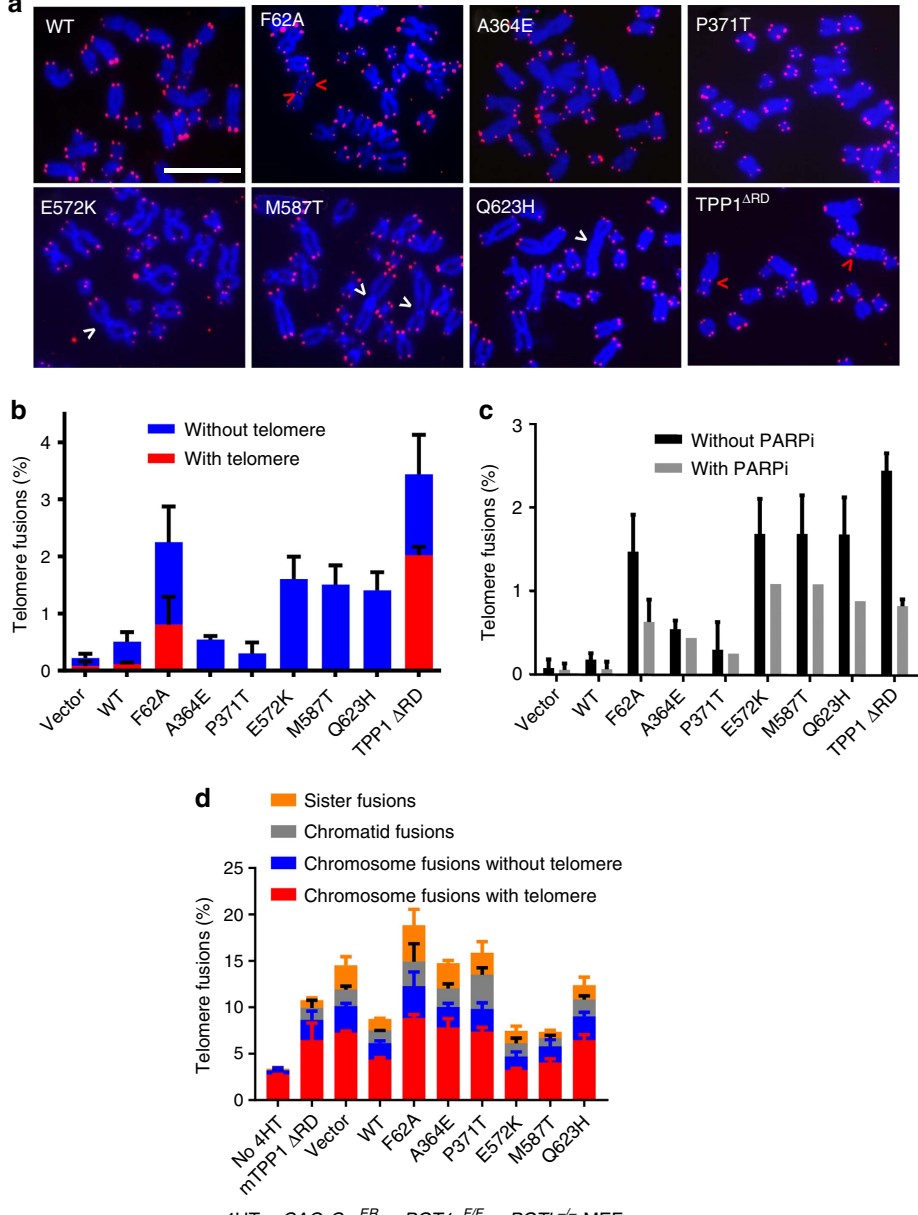

**Figure 7 | POT1 C-terminal mutations promote A-NHEJ-mediated repair.** (**a**) Chromosome fusions in IMR90 cells infected with WT POT1 or POT1 mutants. Metaphase spreads were analysed by peptide nucleic acid - fluorescence *in situ* hybridization (PNA-FISH). Arrowheads point to fusion sites with (red) or without (white) telomere signals. Scale bar, 25 μm. (**b**) Quantification of percentage of telomere fusions in **a**. A minimum of 30 metaphase data were scored per experiment. Error bar (s.e.m.) was derived from three independent experiments. (**c**) Quantification of the percentage of telomere fusions in immortalized IMR90 infected with WT POT1 or POT1 mutants in the presence or absence of the PARP inhibitor PJ34. A minimum of 100 nuclei were scored per experiment. Error bars represent the s.e.m. from three independent experiments. (**d**) Quantification of the percentage of telomere fusions in *CAG-Cre^{ER}*; *mPOT1a^{F/F}*, *mPOT1b^{−/−}* MEFs reconstituted with the indicated DNA constructs and then treated with 4-HT. Error bars represent the s.e.m. from three independent experiments with a minimum of 30 metaphases scored per experiment.

caused by POT1C mutations in human cancer. The $POT1_{OB3-HJRL}$–$TPP1_{PBM}$ complex structure reveals that the $POT1^{E572K}$ and $POT1^{M587T}$ mutations do not disrupt interaction with TPP1, affect POT1 stability nor telomeric localization (Fig. 5b–d). $POT1^{Glu572}$ is on the surface of POT1 away from the POT1–TPP1 interface and makes no contact with the rest of POT1 (Supplementary Fig. 12a). Therefore, $POT1^{E572K}$ is not expected to impact POT1–TPP1 interaction, and this notion is supported biochemically (Fig. 5b). Although $POT1^{Met587}$ is buried inside $POT1_{OB3}$, structural modelling analysis suggests that the structural perturbation caused by the M587T mutation is tolerable for POT1 folding, stability and

interaction with TPP1; this is also demonstrated biochemically (Fig. 5b and Supplementary Figs 10 and 12b). While it is as yet unclear how these mutations impact on the telomere end protective functions of POT1, they do induce telomere dysfunction in WT cells in a dominant-negative manner (Figs 6a,b and 7a–d). In contrast to these two mutations, $POT1^{A364E}$ and $POT1^{P371T}$ affect protein stability, resulting in very low steady-state protein levels (Supplementary Fig. 10). The side chain of $POT1^{Ala364}$ is buried inside $POT1_{OB3}$ and the long acidic side chain of the $POT1^{A364E}$ mutation is not compatible with the hydrophobic local environment (Supplementary Fig. 12c), resulting in interference with the

folding of POT1, yielding an unstable protein. Similarly, the cyclized side chain of POT1$^{Pro371}$ defines the local kink conformation within strand β3 of POT1$_{OB3}$ (Supplementary Fig. 12d), and a threonine mutation at this position would lose this key structural function and lead to unstable folding of POT1. In addition, the side chain of POT1$^{Pro371}$ also stacks with the phenol ring of TPP1$^{Tyr306}$, contributing to the interaction with TPP1 (Supplementary Fig. 12d). *In vitro* gel filtration analysis supported these observations, and showed that POT1$^{A364E}$ and POT1$^{P371T}$ mutant proteins formed protein aggregates, while other POT1 mutants (E572K, M587T and Q623H) generated correctly folded products (Supplementary Fig. 13a). These structural and biochemical analyses are consistent with the fact that even co-expression of TPP1 could not increase the reduced protein levels of POT1$^{A364E}$ and POT1$^{P371T}$ (Fig. 5b and Supplementary Fig. 10). POT1$^{Gln623}$ is located at the surface of POT1$_{OB3}$ and its side chain mediates two hydrogen-bonding interactions with the main chain of TPP1$^{Cys314}$ (Fig. 4f). Although POT1$^{Q623H}$ only destabilizes the third POT1–TPP1 interaction module and does not completely abolish the POT1–TPP1 interaction, this mutant is unstable relative to WT POT1 (Fig. 5b and Supplementary Figs 10, 12a and 13b). All three unstable mutations (A364E, P371T and Q623H) are unable to completely protect telomeres from activating a DDR nor reduce aberrant repair, resulting in the generation of chromosomal fusions and genome instability that are cancer promoting (Figs 6c,d and 7a–d and Supplementary Fig. 11a).

## Discussion

Shelterin proteins POT1 and TPP1 form a stable heterodimer that protects chromosome ends from engaging in a DDR, prevents aberrant repair by A-NHEJ and regulates telomerase-mediated telomere extension[51]. Previous studies established that POT1 and TPP1 are the mammalian homologues of ciliated protozoan *O. nova* TEBP α and β proteins[15,24,27]. In this work, we provide further structural evidence of the resemblance between human POT1 and *O. nova* TEBPα: like TEBPα, POT1 also contains a third OB fold that mediates interaction with TPP1 (Fig. 1c). However, POT1 contains a unique HJRL domain that inserts within POT1$_{OB3}$ and also contributes to the interaction with TPP1 (Fig. 1c). Strikingly, our XL-MS and SAXS data suggest that instead of a heart-shaped TEBPα–β-like structure, the POT1–TPP1 complex adopts an elongated V-shaped conformation with POT1$_{OB1-OB2}$ and TPP1$_{OB}$ at opposite sides of the complex (Fig. 3g).

The *O. nova* telomeric 3′ overhang is very short and a single heart-shaped TEBPα–β complex sandwiches the overhang in the middle to sequester the 3′ end[27]. Notably, human chromosomes end in a long ss overhang with a length that could be several hundred nucleotides long[3,52]. If the POT1–TPP1 complex adopts a similar heart-shaped architecture as that of TEBPα–β, it would only be able to bind the very end of the 3′ overhang. In contrast, the extended V-shaped architecture of POT1–TPP1 is suited for binding not only to the 3′ end of the overhang but also to internal ss telomeric regions. Thus, the entire ss overhang could be coated with POT1–TPP1 complexes. In addition, human telomeres can form a large loop structure called a t-loop, with the ss terminus paired with an internal region[53]. Formation of a t-loop displaces an internal segment of ss telomeric repeats to form a D-loop[54]. It is possible that this ss internal segment also associates with and is protected by the POT1–TPP1 complex. Therefore, although the *O. nova* TEBPα–β and human POT1–TPP1 complexes share many same domains, it is likely that during evolution the architectures of these complexes are rearranged to meet different functional needs of the two organisms.

Another insight from our studies is the recruitment of telomerase by TPP1 to telomeres. Our previous results revealed

that when bound to internal telomeric ss sequences with a free 3′ overhang, the POT1–TPP1 complex can stimulate telomerase activity and processivity[15]. Later studies showed that the TEL patch on the OB fold of TPP1 mediates the direct interaction with TERT and plays a key role in both telomerase recruitment and telomerase activity stimulation[15,16,19,55]. In our POT1–TPP1 complex model, TPP1$_{OB}$ is located at one end of the V-shaped envelope away from the POT1 ssDNA-binding site (Fig. 3g). TPP1$_{OB}$ in this position may possess the optimal conformation for the interaction with TERT. It is also possible that this extended conformation of POT1–TPP1 allows telomerase to access the DNA terminus and stabilizes the association between telomerase and telomere substrate during the translocation step of telomere extension. Further structural studies are needed to understand the molecular mechanism of how POT1–TPP1 mediates telomerase recruitment and regulates its activity.

Finally, our results reveal that the POT1 OB3 and HJRL domains promote POT1 stability through interaction with TPP1. Somatic POT1C TNBC mutations cluster preferentially within OB3, some of which function as dominant negatives to activate a DDR and initiate A-NHEJ-mediated chromosome fusions. POT1C OB3 mutations that specifically disrupt the POT1–TPP1 interaction destabilize POT1, reducing its accumulation on telomeres. We have previously shown that POT1 is required to repress the initiation of an ATR-replication protein A (ATR-RPA) damage signalling pathway and the generation of 3′ ss overhangs essential to promote A-NHEJ-mediated repair[22,23,51]. A-NHEJ is a microhomology-based error-prone repair pathway that fosters the generation of gross chromosomal abnormalities and has emerged as a primary DNA repair pathway in many human cancers. While the POT1 N-terminal OB folds were thought to be solely responsible to prevent RPA from gaining inappropriate access to telomeric ssDNA, our data reveal that the POT1 C-terminal OB3 is also important in this process. It appears that human tumours utilize two approaches to disrupt POT1 function: either by mutating the POT1N OB folds to eliminate its ability to bind to telomeric ssDNA and thereby activate a DDR at telomeres, or by mutating the POT1–TPP1-binding interface to destabilize POT1. Whatever the mechanism, loss of POT1 function promotes end-to-end chromosome fusions and cancer formation, suggesting that POT1 is critically important for preventing the onset of tumour promoting genomic instability.

## Methods

**Cell lines and treatments.** IMR90, U2OS cells and *CAG-Cre$^{ER}$; mPot1a$^{F/F}$, mPOT1b$^{-/-}$* MEFs were cultured in DMEM supplemented with 10% FBS and maintained in 5% $CO_2$ at 37 °C. For retroviral infections, DNA constructs were transfected into 293T cells using Fugene 6 and packaged into viral particles. Viral supernatant was collected 48–72 h after transfection, filtered through 0.45-µm pore-size membrane and directly used to infect cells. Robust expression of WT or mutant human POT1 constructs in *CAG-Cre$^{ER}$; mPOT1a$^{F/F}$, mPOT1b$^{-/-}$* MEFs was accomplished by co-expression with human TPP1. After reconstitution with POT1 mutants, *CAG-Cre$^{ER}$; mPot1a$^{F/F}$, mPOT1b$^{-/-}$* MEFs were treated with 1 µM of 4-HT for 48 h to delete endogenous mPOT1a. 0.5 µg ml$^{-1}$ of PARP1 inhibitor PJ34 (CaliBiochem) was used to inhibit PARP1 activity.

**Western analyses.** The antibodies used for western blot analysis were as follows: anti-γ-H2AX (Millipore 05–636, 1:1,000 dilution); anti-Flag (Sigma F3165, 1:1,500 dilution), anti-HA (Sigma H3663, 1:1,000 dilution), anti-γ-tubulin (Sigma T6557, 1:1,000 dilution), anti-Myc (Millipore #05–724, 1:2,000 dilution), anti-Myc antibody (Santa Cruz sc-789, 1:1,000 dilution) and anti-GFP (Santa Cruz sc9996, 1:1,000 dilution). Human POT1 and point mutations were generated by site-directed mutagenesis in retrovirus expression vectors, MSCV-IRES-GFP or MSCV-IRES-puro.

**Co-immunoprecipitation.** Co-IP of POT1 and TPP1 was performed in human 293T cells using transient transfection. 293T cells ($4 \times 10^6$ cells) were plated in

a 6 cm-dish and transfected with X-tremeGENE HP DNA Transfection Reagent (Roche) and indicated plasmid DNAs following the manufacturer's recommendation. Five microgram $3 \times$ Flag WT POT1 and $3\,\mu g$ Myc-TPP1 mutants (or $5\,\mu g$ Myc-POT1 mutants and $3\,\mu g$ $3 \times$ Flag WT TPP1) were added per transfection. After 48 h of transfection, cells were washed with PBS, resuspended with $500\,\mu l$ lysis buffer (50 mM Tris-HCl, pH 7.4, 150 mM NaCl, 0.5% Triton X-100, 1 mM EDTA, complete protease inhibitor cocktail (Roche)), sonicated at 70 W for 5 s $\times$ 12 times and spun at $15,000g$ for 10 min at 4 °C. Supernatants were collected and used directly for IP. For co-IP of $3 \times$ Flag-POT1 WT and Myc-TPP1 mutants, $8\,\mu l$ mouse anti-Myc antibody (Santa Cruz, sc-40) was mixed with supernatants overnight and $15\,\mu l$ protein G-agarose beads were added during the final hour. For co-IP of Myc-POT1 mutants and $3 \times$ Flag-TPP1 WT, $10\,\mu l$ M2 beads (Sigma) were mixed with supernatants overnight. Beads were washed with lysis buffer for three times and analysed by SDS–PAGE. The $3 \times$ Flag WT POT1 and $3 \times$ Flag WT TPP1 were detected with mouse anti-Flag antibody and the Myc-TPP1 mutants and Myc-POT1 mutants were detected with rabbit anti-Myc antibody.

**DNA binding assay.** To examine whether mutant human POT1 was able to bind to ss telomeric DNA in vitro, 293T cells expressing WT or mutant HA–POT1 were lysed and incubated in $TEB_{150}$ buffer (50 mM Hepes pH 7.3, 150 mM NaCl, 2 mM $MgCl_2$, 5 mM EGTA, 0.5% Triton X-100, 10% Glycerol, proteinase inhibitors) overnight at 4 °C with streptavidin–sepharose beads (Invitrogen) coated with biotin-Tel-G $(TTAGGG)_6$ oligo. Bound complexes were washed three times with the same buffer.

**CHX chase experiment.** To examine the stability of WT POT1 and POT1 mutants after inhibition of protein synthesis, $30\,\mu g\,ml^{-1}$ of CHX (Sigma #C4859) was added to 293T cells co-transfected with human TPP1 and either WT POT1 or POT1 mutants. After 48 h, the cells were collected at different time points, lysed in urea lysis buffer (8 M urea, 50 mM Tris-HCl, pH 7.4 and 150 mM β-mercaptoethanol) and proteins examined by western analysis.

**Cytogenetic analyses.** Cells were treated with $0.5\,\mu g\,ml^{-1}$ of colcemid for 4 h before collection. Trypsinized cells were treated with 0.06 M KCl, fixed with methanol:acetic acid (3:1) and spread on glass slides. Metaphase spreads were hybridized with $5'$-Cy3-OO-$(CCCTAA)_4$-$3'$ probe. For CO-FISH, cells were treated with BrdU for 14 h before addition of colcemid. Metaphase spreads were sequentially hybridized with $5'$-FAM-OO-$(TTAGGG)_4$-$3'$ and $5'$-Cy3-OO-$(CCCTAA)_4$ probes. For the TIF assay, cells were seeded in eight-well chambers and immunostained with anti-γ-H2AX primary antibody and FITC-conjugated secondary antibody, then hybridized with the $5'$-Cy3-OO-$(CCCTAA)_4$-$3'$ probe to visualize both DNA damage foci and telomeres. Telomere fusions are scored as: total number of chromosome fusions divided by the total number of chromosomes $\times$ 100%. A minimum of 30 metaphases were scored per experiment, with three independent experiments performed.

**Protein expression and purification.** Human POT1C (residues 320–634) and $TPP1_{PBM}$ (residues 266–320) were cloned into a modified pET28a vector with a SUMO protein fused at the N terminus after the $6 \times$ His tag and pMAL-C2X vector with an MBP protein fused at the N terminus, respectively. The POT1C–$TPP1_{PBM}$ complex was co-expressed in E. coli BL21(DE3). After induction for 20 h with 0.1 mM IPTG at 23 °C, the cells were collected by centrifugation, and the pellets were resuspended in lysis buffer (50 mM Tris-HCl, pH 8.0, 150 mM NaCl, 10% glycerol, 1 mM PMSF, 5 mM benzamidine, $1\,\mu g\,ml^{-1}$ leupeptin and $1\,\mu g\,ml^{-1}$ pepstatin). The cells were then lysed by sonication, and the debris was removed by ultracentrifugation. The supernatant was mixed with Ni-NTA agarose beads (Qiagen) and rocked for 2 h at 4 °C before elution with 250 mM imidazole. Then ULP1 protease was added to remove the His-SUMO tag while the protein was incubated with Amylose resin (New England Biolabs) at 4 °C overnight. The protein was eluted with 15 mM maltose (Sigma) and then PreScission protease was added to remove the MBP tag. The protein was further purified by Mono-Q and by gel-filtration chromatography equilibrated with 25 mM Tris-HCl pH 8.0, 150 mM NaCl. The purified protein was concentrated to 30 mg ml$^{-1}$ and stored at $-80$ °C.

Human full-length POT1 was cloned into a modified Bac-to-Bac vector containing an N-terminal glutathione S-transferase, preceding the multiple cloning sites (Invitrogen). Human TPP1N was also cloned into Bac-to-Bac vector with a $6 \times$ His tag at the N terminus. For POT1–TPP1N complex expression, High five insect cells were infected at $\sim 3 \times 10^6$ cells ml$^{-1}$ with a multiplicity of infection of 10 plaque-forming unit ml$^{-1}$ recombinant baculovirus. The cells were collected after 68 h by centrifugation. The pellets were resuspended in lysis buffer (50 mM Tris-HCl, pH 8.0, 150 mM NaCl, 10% glycerol, 1 mM PMSF, 5 mM benzamidine, $1\,\mu g\,ml^{-1}$ leupeptin and $1\,\mu g\,ml^{-1}$ pepstatin). The cells were then lysed by sonication, and the debris was removed by ultracentrifugation. The supernatant was mixed with Ni-NTA agarose beads (Qiagen) and rocked for 2 h at 4 C° before elution with 250 mM imidazole. The protein was then mixed with glutathione Sepharose-4B beads (GE healthcare) and rocked overnight at 4 °C and eluted with 15 mM reduced glutathione (Sigma). PreScission protease was then added to remove the N-terminal $6 \times$ His and glutathione S-transferase tags. The protein was further purified by Mono-Q and gel-filtration chromatography equilibrated with

25 mM Tris-HCl pH 8.0, and 150 mM NaCl. The purified proteins were concentrated to 10–15 mg ml$^{-1}$ and stored at $-80$ °C.

**Crystallization and structure determination.** Crystals of the native POT1C–$TPP1_{PBM}$ complex were grown by sitting-drop vapour diffusion at 4 °C. The precipitant well solution consisted of 2.0 M $NaH_2PO_4/K_2HPO_4$ (2:3), 0.2 M sodium citrate, 0.1 M acetate pH 4.4. Crystals were gradually transferred into a collecting solution containing 2.6 M $NaH_2PO_4/K_2HPO_4$ (2:3), 0.2 M sodium citrate, 0.1 M acetate pH 4.4, 25% glycerol, followed by flash-freezing in liquid nitrogen for storage. Crystals of SeMet-labelled POT1C–$TPP1_{PBM}$ were grown in the similar condition. Data sets were collected under cryogenic conditions (100 K) at the Shanghai Synchrotron Radiation Facility (SSRF) beamlines BL18U1 and BL19U1. A 3.1-Å SeMet-SAD dataset of POT1C–$TPP1_{PBM}$ was collected at the Se peak wavelength (0.97876 Å) and was processed by HKL3000 (ref. 56). Ten selenium atoms were located and refined, and the single-wavelength anomalous diffraction data phases were calculated using XDS[57]. The initial SAD map was substantially improved by solvent flattening. The model was then refined using a native dataset with a 2.1-Å resolution using Phenix[58], together with manual building in Coot[59]. In the final Ramachandran plot, the favoured and allowed residues are 99.1 and 100.0%, respectively. All the crystal structural figures were generated using PyMOL[60].

**Yeast two-hybrid assay.** The yeast two-hybrid assays were performed using L40 strains harbouring pBTM116 and pACT2 (Clonetech) fusion plasmids. The colonies containing both plasmids were selected on -Leu -Trp plates. The β-galactosidase activities were measured by liquid assay[61].

**Crosslinking and mass spectrometry analysis.** XL-MS is a valuable tool for providing information about protein folding and protein–protein interaction without high-resolution structures. Although XL-MS cannot compete at the level of details and global information provided by traditional high-resolution methods like X-ray crystallography, NMR or Cryo-EM, it is more tolerable in term of sample concentration and purity. Additionally, XL-MS can be conducted in vitro under the condition that mimic native protein environment and capture interactions from dynamic states. XL-MS studies involve protein crosslinking through a chemical linker, digestion of the crosslinked protein complex into peptides and identification of the crosslinked peptides, consequently, proximal residue pairs.

The buffer of the purified POT1–TPP1N–T10 complex was changed to 25 mM Hepes, pH 8.0, 150 mM NaCl at a concentration of 0.5 mg ml$^{-1}$. Complexes were then crosslinked by disuccinimidyl suberate (Thermo Fisher) at a ratio of 1:0.5 (wt/wt). Ammonium bicarbonate (Sigma) with a final concentration of 20 mM was added to terminate the reaction after incubation at room temperature for 1 h. Then SDS–PAGE was used to separate the crosslinked protein complexes. After in-gel digestion with trypsin, peptides were desalted with Pierce C18 spin column (Thermo Fisher) and separated in a home-packed C18 column (75 μm × 15 cm, Phenomenex Aqua 3 μm, 125 Å, C18 beads) using a proxeon EASY-nLC liquid chromatography system by applying a stepwise gradient of 0–85% acetonitrile in 0.1% formic acid. Peptides eluted from the LC column were directly electrosprayed into the mass spectrometer with a distal 2 kV spray voltage. Data-dependent tandem mass spectrometry (MS/MS) analysis was performed on Thermo Q-Exactive instrument (Thermo Fisher, San Jose, CA) in a 60-min gradient. Raw data were processed with pLink software[62].

**SAXS analysis and ab initio 3D shape reconstructions.** SAXS is a technique for low-resolution structural characterization of biological macromolecules in solution. SAXS can be used to probe proteins, nucleic acids and their complexes without the need of crystallization and without the molecular weight limitation inherent in other high-resolution methods such as NMR or Cryo-EM. Being complementary to the high-resolution methods, SAXS is often used together with them. The technique provides several key parameters of a biological macromolecule such as the molecular weight MW, radius of gyration $R_g$, maximum intramolecular distance $D_{max}$ and overall 3D structures.

SAXS experiments were performed at beamline BL19U2 of National Center for Protein Science Shanghai (NCPSS) at SSRF. The wavelength λ of X-ray radiation was set as 1.033 Å. Scattered X-ray intensities were collected using a Pilatus 1 M detector (DECTRIS Ltd). The sample-to-detector distance was set such that the detecting range of momentum transfer ($q = 4p$ sinq/l, where 2q is the scattering angle) of SAXS experiments was 0.01–0.45 Å$^{-1}$. To reduce the radiation damage, a flow cell made of cylindrical quartz capillary with a diameter of 1.5 mm and a wall of 10 μm was used. SAXS data were collected as $20 \times 1$ s exposures and scattering profiles for 20 passes were compared at 10 °C using 60 μl sample in 25 mM Tris-HCl, pH 8.0, 150 mM NaCl and 5 mM DTT. Measurements were carried out at three different concentrations 2.5, 5 and 10 mg ml$^{-1}$. The 2D scattering images were converted to 1D SAXS curves through azimuthally averaging after solid angle correction and then normalized with the intensity of the transmitted X-ray beam, using the software package BioXTAS RAW[63]. Background scattering was subtracted using PRIMUS[64] in the ATSAS software package[65]. Linear Guinier plots in the Guinier region ($q \times R_g < 1.3$) were confirmed. Pair distance distribution functions of the particles P(r) and the maximum sizes $D_{max}$ were computed using

GNOM[66] and molecular weights were estimated by using the SAXS MoW2 package[67]. The *ab initio* shapes of the complexes were determined using DAMMIF[44] with 20 runs for each experimental group, and DAMAVER[68] was used to analyse the normalized spatial discrepancy between the 20 models. The filtered SAXS model by DAMFILT was showed in UCSF Chimera[69]. Docking of crystal structures into the SAXS envelopes was performed using PyMOL[60].

**EMSA.** The following oligonucleotides were synthesized and purified in the EMSA assay.

Oligo 1 (68-mer): 5′-CAGATGGACATCTTTGCCCACGTTGACCCGAA (TTAGGG)₄CCATGGTAGCCC-3′, Oligo 2 (62-mer): 5′-GGGCTACCATGG (CCCTAA)₄TTGACATGCTGTCTAGAGACTATCGC-3′, Oligo 3 (62-mer): 5′-GCGATAGTCTCTAGACAGCATGTCCG(TTAGGG)₄CAAGCGTCCGAG-3′, Oligo 4 (68-mer): 5′-CTCGGACGCTTG(CCCTAA)₄CGCGGGTCAACGTGGG CAAAGATGTCCATCTG-3′. Oligo 1 was labelled with 6-FAM at 5′ terminus. The annealing reaction contains 200 nM oligo 1 and 1 μM each of oligo 2, oligo 3 and oligo 4. Holliday junctions were assembled by annealing 4 oligos at 95 °C for 5 min, then 65 °C for 10 min, then 37 °C for 10 min, and finally gradually cooled down from 37 °C to room temperature[70]. The binding mixture contained 20 nM Holliday junction substrate and 0–64 μM of purified POT1C–TPP1_PBM complex.

**Melting curve measurement.** Melting curves were taken on a ViiA 7 Real-Time PCR System (Thermo Fisher Scientific) using 20 μl reaction volume containing 0.1 mg ml⁻¹ WT and mutant POT1 proteins and 5xSYPRO Orange Protein Gel Stain (Thermo Fisher Scientific). The temperature was from 20 to 95 °C.

**Data availability.** Coordinates and structure factor amplitudes have been deposited in the Protein Data Bank with accession code 5H65. The data that support the findings of this study are available from the corresponding author on request.

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

## Acknowledgements

We are grateful to protein expression and purification system and mass spectrometry system at the NCPSS for instrument support and technical assistance. We thank D. Yao and W. Qin from BL18U1 and BL19U1 beamlines at NCPSS and SSRF for help with crystal data collection and processing. We thank N. Li, G. Liu and H. Wu from BL19U2 beamline at NCPSS and SSRF for assistance during SAXS data collection and processing. We thank Zuo X. from beamline 12-ID-B at APS for discussion of SAXS data processing. This work was supported by grants from the Ministry of Science and Technology of China (2013CB910402 to M.L.), the National Natural Science Foundation of China (31330040 and 31525007 to M.L., 31500625 to J.W.), the Strategic Priority Research Program of the Chinese Academy of Sciences (XDB08010201) to M.L. and the Youth Innovation Promotion Association of the Chinese Academy of Sciences to J.W. S.C. is supported by the NCI (RO1 CA129037, R01 CA202816, R21 CA200506 and R21 CA182280) and the CT Department of Public Health (15–002167).

## Author contributions

C.C. was responsible for the structural determination and biochemical experiments; P.G. for all experiments for Figures 5,6 and 7; X.C., S.N., S.S. H.S. and C.F. for making constructs and some of the protein purification; L.W. and R.Z. for crystal data processing; N.L., J.P. and Q.G. for SAXS data processing; M.H. and C.W. for data analysis of XL-MS. J.W., S.C. and M.L. contributed to overall design, interpretation of the studies and manuscript preparation.

## Additional information

**Competing interests:** The authors declare no competing financial interests.

