## [Peer Review File · Nature Communications]

Reviewers' Comments:

Reviewer #1 (Remarks to the Author)

The authors report the crystal structure of a minimal human POT1-TPP1 complex consisting of the C-terminal part of POT1 and the central section of TPP1 - about a quarter of TPP1. The structure reported is important as far as it goes and the structural analysis is well done revealing both the heterodimer interface and the presence of a Holliday Junction Resolvase like domain inserted in the POT1 OB fold 3. In addition, mutational and functional analyses based on mutations found in certain cancers, shows that such mutations disrupt the POT1-TPP1 complex as well as failing to repress DNA damage response. Whilst the structural analysis and some of the biological consequences of mutations are more or less sound, the content of the manuscript is marred by speculations and over interpretations.

p5) How was the POT1C domain used in the structural analyses identified. Does it bind as strongly to TPP1 as the full-length protein? For a solid conclusion of the dimer interface both interaction domains should be characterized - and was not done here. In Figure 1 give aa number defining domains. Some residue No on structure would also be helpful.

p11) and figure 4B – label residues mentioned in text in Figure 4 or it is difficult to follow reasoning when comparing to the TEBP α /TEBP β structure. Also label crosslinked residues POT1 Lys 433 and TPP1Lys 232 in this figure rather than in supplementary.

p12-13) The fitting of domains into the SAXS model envelope is difficult to see – most of domains look blue. More seriously, the fitting is speculative. Proof that the envelope is correctly fitted is to repeat SAXS analysis with a TPP1 construct lacking TPP1 OB1. Also how was the C-terminal part (almost half of TPP1) fitted into envelope – authors have no structural information for this as far as I can tell. I do not think that the model proposed for full length POT1/TPP1 adds much to knowledge as it is highly speculative.

p14) It is stated that most POT1 mutants expressed in cells are unstable. Not clear to me what this means. How is unstable defined? They simply do not detect POT1. The reason for this could equally well be protein precipitation. Normally to define unstable one would express the mutant protein and do a melting curve and compare to the wt protein.

p14-15) Figure 5 showing confocal data is poor – unable to see what is going on.

P19) The Discussion section is far too speculative – certainly not warranted by the structural data presented – and based on a model of the full length POT/TPP1 dimer that itself is far from proven. Discussion should be cut down and stick to the structure and interpretation of mutations based on the real structural data presented.

Reviewer #2 (Remarks to the Author)

In mammals, it is known that POT1 and TPP1 form a heterodimer to protect telomere overhangs. Mutations of POT1 have been found in a variety of human cancers. The manuscript by Chen and colleagues reported the crystal structure of POT1-TPP1 interaction domains POT1C and TPP1 PBM. Such structural information is important to understanding the molecular and pathological basis of POT1 mutants in human cancer. The authors further characterized several human POT1 mutants and found that some of these mutants disrupted POT1-TPP1 binding, while others impaired DNA protection function of POT1-TPP1 complex. Overall, the manuscript provides structural insight into the mechanism of POT1-TPP1 mediated telomere overhang protection and how POT1 mutations may induce genomic stability and lead to cancer development. However, several points need to be addressed before acceptance for publication.

1. The authors analyzed several POT1 mutants found in human cancers and claimed that “We identified several missense mutations in human cancers that disrupt the POT1C-TPP1 interaction, resulting in POT1 instability.” However, their data indicate that these mutations do not affect POT1 interaction with TPP1 at all. A364 and P371 in POT1 seemed to greatly influence stability, and therefore not the best mutants for this study. The conclusion is also conflicting for the most relevant mutant Q623H. Line 411 said “Q623H mutations disrupted the interaction between TPP1 nor affected the telomeric localization of POT1”, while 422 said “the Q623H mutation only destabilizes the third POT1-TPP1 interaction module and does not completely abolish the POT1-TPP1 interaction “. It is necessary to revise their conclusions and models according to the data.

2. Experiments in Figs.4 and Figs.5 are problematic in terms of revealing mechanistic insight. The first part showed an amazing structure of POT1C with TPP1 PBM, and then the second part showed the impact of POT1C mutations on telomeres, but the mechanisms of action for these mutants are pretty much unknown. Are there mutations on POT1C that can affect the interaction with TPP1? Are there amino acids on POT1C, identified from the structure as Fig2 did, that are important for the interaction with TPP1 and can impact telomere function and genomic stability?

3. In addition to PTM, the authors used TPP1 RD mutants in their experiments. The authors should better explain the RD domain. What is the relationship between RD and PBM? The RD domain appears just slightly larger than PBM.

4. The authors utilized different techniques (SASX and XL-MS etc.) to study POT1-TPP1 interaction. It would be beneficial to the general audience to explain clearly the rationale for choosing these approaches as well as their pros and cons. The methods section needs to be improved.
5. The authors should also check their references to make sure the correct papers are referenced in the proper places. For instance, line 330, “localization of POT1 to telomere requires TPP1”, should reference original studies.
6. In Fig 2g, TPP1 double mutant in the last lane has two bands, why? Fig 2g lacks a negative control. In Fig 2h, the input of lane 3 is too low, for both Myc-TPP1 and Flag-POT1, as was the Flag-POT1 input for lane 2. It made it difficult to interpret the IP data. The co-IP experiment should be repeated with proper controls and better data.
7. Why was POT1 A364E used in Fig. 5b, but POT1 A364K used in the later experiments? In Fig. 5e, input is low for both POT1 A364K and P371T compared to others. It is therefore difficult to conclude that the binding was at reduced levels. Why did Fig 6 not include mutants A 264E and P371T? The data should be consistent and complete as in Fig 5.
8. Line 140, the EMSA data of POT1HJRL with Holliday junction should be moved to supplemental figures.

Reviewer #3 (Remarks to the Author)

This manuscript describes the structure of C-terminal domain of the telomere protein POT1 complexed with the POT1-binding domain of its partner protein TPP1. The structure is important because it provides much-needed insight into how POT1-TPP1 can bind ssDNA and at the same time recruit telomerase to the telomere. The structure is particularly intriguing because it demonstrates that the structural components of the ssDNA telomere end-binding complex are conserved from humans to ciliates but that the overall three dimensional architecture of the complex has evolved. Presumably the evolution reflects the differences in telomere structure and telomerase composition hence the different needs in terms of telomere protection and telomerase recruitment/stimulation. The structural work is well done with a nice meshing of various biophysical techniques (X-ray crystallography, cross-linking and mass spec, and SAX analysis) to generate a sophisticated model of the POT1-TPP1 structure.

The second part of the manuscript describes some previously unknown cancer-associated mutations in POT1 that lie in the C-terminal region that is the topic of this study. The authors

demonstrate that mutant POT1 proteins have a dominant negative effect that impacts the ability of the POT1-TPP1 complex to prevent the telomere from eliciting a DNA damage response and DNA repair (Alt-NHEJ). The cancer link provides added interest to the structural study, although ultimately the actual mechanistic cause(s) of the resulting deficiencies in telomere maintenance remain unclear.

Overall this is a strong piece of work that will be of quite broad interest. The following editing and revisions are needed before it is ready for publications.

Introduction: The authors need to mention that the TPP1 OB fold is the domain that interacts with telomerase and they should provide a brief discussion of how TPP1 modulates telomerase activity as this information is needed later (see comments below about the discussion section).

The introduction needs some editing. Page 3, Lines 43-45: something is missing, perhaps the word, together. Lines 55-58: the difference between the functions of the two mouse POT1 proteins and the single human POT1 is not made clear. Page 4 line 59, “protozoan” is missing, line 61, remove the capital T.

Figure 1a. The authors need to label the telomerase binding domain in TPP1 and the legend (or the figure) should make it clear that PBM stands for POT1 binding domain and TBP is TIN2 binding domain. Also, it would be helpful to indicate the residue numbers for the amino acids that delineate the various domains within TPP1 and POT1.

Page 8, line 161: Mention that TPP1 residues 271, 275 and 279 are in helix H1.

Page 10 line 209: change rooms to room. Line 212: add “the” to read the three strands....

Page 14 It would be helpful to explain the logic of using the F62A and Q623H mutations in the main text. Currently, this information is buried in the legend for Fig. 5.

Page 16 line 361 and Fig. 6 b-d. The terminology used to describe the frequency of chromosome fusions (Total fusion per chromosome or % fusions per chromosome) is most confusing and cannot be correct. Do the authors mean “number of fusions per 100 chromosomes? The authors must correct their terminology so that the frequency of fusions is correct and can be compared to fusion frequency reported for other mutants.

The experiment with SNM1b/Apollo^{-/-} MEFs (Fig 6d and all text relating to this figure) should be removed from the manuscript as it does not add useful information. The sudden switch to mouse mutants is unwarranted because (i) the mutants relevant to this manuscript (E572K and M587T don't have much effect on the fusion rate, (ii) it remains to be demonstrated that Apollo

also processes leading strand telomeres in human cells and (iii) mouse Pot1 proteins are different from the single human POT1 protein so it is unclear how the Pot1a/b mutants relate to the human POT1 mutants.

Page 19 line 438: add the word protozoan to read “ciliated protozoan *O. nova*”

Page 20, line 461: add the word is to read “and is protected by”

The discussion does not pack as much punch as it could. Additional discussion of how the POT1/TPP2pbd structure might impact telomerase action could strengthen it. Personally, I find it intriguing that the *O. nova* and human POT1/TPP1 complex contain so many of the same domains but they are re-arranged to deal with the different needs of the two organisms. The authors point out that human POT1/TPP1 needs to bind along the length of the 3' overhang. It would be helpful to point out that this not the case for *O. nova* where the overhangs are very short so they are likely bound by a single complex which is tailored to sequester the 3' end. It would also be helpful to mark the TEL-patch on TPP1 in Fig. 4G and to then point out how this appears to place the telomerase interaction site on a long arm that is distant from the POT1 ssDNA binding site. Perhaps this is what allows telomerase to access the DNA terminus and translocate as it adds repeats (see the Cech lab paper showing that a single POT1-TPP1 bound distant from the DNA 3' end can stimulate telomerase activity). I understand that the TIN2-binding domain of TPP1 is missing but it looks like this would sit closer to POT1 OB1 and OB2.

Point-to-point responses to reviewers' comments

Structural insights into POT1-TPP1 interaction and POT1 C-terminal mutations in human cancer

NCOMMS-16-18882

We are grateful that the reviewers found this manuscript “a strong piece of work that will be of quite broad interest”, with “the structural analysis well done, providing important structural insight into the mechanism of POT1-TPP1 mediated telomere overhang protection and how POT1 mutations may induce genomic stability and lead to cancer development”.

Below is our point-by-point answer to the reviewers' comments.

Reviewer #1:

p5) How was the POT1C domain used in the structural analyses identified. Does it bind as strongly to TPP1 as the full-length protein? For a solid conclusion of the dimer interface both interaction domains should be characterized - and was not done here.

In our previous study (Lei et al 2004), we determined the crystal structure of POT1 N-terminal two OB folds (residues 1-300) complexed with a 10 nt telomere DNA. Secondary structure prediction analysis shows that there is a short unstructured loop (residues 300-320) between the N-terminal OB folds and the C-terminal domain of POT1 (POT1C). We used yeast two-hybrid analysis to show that POT1C interacts with TPP1 and POT1 N-terminal OB folds are not required for the interaction between POT1 and TPP1. This data is now shown in Supplementary Figure 1a in the revised manuscript.

In Figure 1 give aa number defining domains. Some residue No on structure would also be helpful.

Thank you for this good point. We added residue numbers in Figure 1a to define the boundaries of various domains of POT1 and TPP1. The N- and C-termini are labeled to show the beginning and the end of the protein domains used in our structural study.

p11) and figure 4B – label residues mentioned in text in Figure 4 or it is difficult to follow reasoning when comparing to the TEBP α /TEBP β structure. Also label crosslinked residues POT1 Lys 433 and TPP1Lys 232 in this figure rather than in supplementary.

We labeled the residue numbers mentioned in text in revised Figure 3b. We also labeled the crosslinked residues POT1 Lys 433 and TPP1 Lys 232 in revised Figure 3b.

p12-13) The fitting of domains into the SAXS model envelope is difficult to see – most of domains look blue. More seriously, the fitting is speculative. Proof that the envelope is correctly fitted is to repeat SAXS analysis with a TPP1 construct lacking TPP1 OB1.

We thank this reviewer for raising this important point. We followed this reviewer's suggestion and carried out a SAXS analysis with full-length POT1 and a TPP1 construct without TPP1 OB

(residues 260-334). The derived envelope demonstrates that the POT1-TPP1_{PBM} complex adopts a V-shaped topology and the two arms of the envelope have roughly equal length (Supplementary Fig. 6e). This envelope can be nicely superposed onto the envelope of the V-shaped POT1-TPP1N complex with a correlation of 0.945, except that one arm of the POT1-TPP1N complex is substantially longer than the one in the POT1-TPP1_{PBM} complex (Supplementary Fig. 6f). Notably, TPP1 OB in the POT1-TPP1N complex model fits just in the difference between the two envelopes -- the tip of the long arm of the POT1-TPP1N complex (Supplementary Figs. 6f and 6g). This clearly shows that our model of the POT1-TPP1N complex is correctly fitted in the envelope (Fig. 3g). We have included these important points into the paper.

Also how was the C-terminal part (almost half of TPP1) fitted into envelope – authors have no structural information for this as far as I can tell. I do not think that the model proposed for full length POT1/TPP1 adds much to knowledge as it is highly speculative.

The TPP1 construct used in this work (TPP1N, residues 89-344) is not full-length TPP1. It lacks the C-terminus. So the model we proposed is not for full-length POT1-TPP1 but rather for POT1-TPP1N. We now make this point clear in the revised manuscript.

p14) It is stated that most POT1 mutants expressed in cells are unstable. Not clear to me what this means. How is unstable defined? They simply do not detect POT1. The reason for this could equally well be protein precipitation. Normally to define unstable one would express the mutant protein and do a melting curve and compare to the wt protein.

We thank this reviewer for raising these important points. We now provide new data from a cycloheximide chase assay (Supplementary Fig. 9) demonstrating that POT1 C-terminal mutants A364E and P371T exhibit markedly reduced levels even in the presence of TPP1. Gel filtration analysis show that these POT1C mutants form aggregates, revealing that both POT1 A364E and P371T mutants interfere with the protein folding (Supplementary Fig. 12a). In contrast, the other POT1C mutations POT1E572K, POT1M587T and POT1Q623H all behave like WT POT1 in gel filtration assays and except for POT1 Q623H have similar melting temperatures as WT POT1 (Supplementary Figs. 12a and 12b). POT1 Q623H has a slightly lower melting temperature, suggesting that this mutation is slightly less stable than WT POT1 (Supplementary Fig. 12b).

p14-15) Figure 5 showing confocal data is poor – unable to see what is going on.

We have enlarged this image to make the data clearer in the revised Figures 5, 6 and 7.

P19) The Discussion section is far too speculative – certainly not warranted by the structural data presented – and based on a model of the full length POT/TPP1 dimer that itself is far from proven. Discussion should be cut down and stick to the structure and interpretation of mutations based on the real structural data presented.

We have now shortened the Discussion section and limited our discussions to the structural data on hand.

Reviewer #2:

1. The authors analyzed several POT1 mutants found in human cancers and claimed that “We identified several missense mutations in human cancers that disrupt the POT1C-TPP1 interaction, resulting in POT1 instability.” However, their data indicate that these mutations do not affect POT1 interaction with TPP1 at all. A364E and P371T in POT1 seemed to greatly influence stability, and therefore not the best mutants for this study. The conclusion is also conflicting for the most relevant mutant Q623H. Line 411 said “Q623H mutations disrupted the interaction between TPP1 nor affected the telomeric localization of POT1”, while 422 said “the Q623H mutation only destabilizes the third POT1-TPP1 interaction module and does not completely abolish the POT1-TPP1 interaction”. It is necessary to revise their conclusions and models according to the data.

We thank this reviewer for raising these important points. We now provide new data from a cycloheximide chase assay (Supplementary Fig. 9) demonstrating that POT1 C-terminal mutants A364E, P371T exhibit markedly reduced levels even in the presence of TPP1. Gel filtration analysis show that these POT1C mutants form aggregates, revealing that both POT1 A364E and P371T mutants interfere with the protein folding (Supplementary Fig. 12a). In contrast, the other POT1C mutations POT1E572K, POT1M587T and POT1Q623H all behave like WT POT1 in gel filtration assays and except for POT1 Q623H have similar melting temperatures as WT POT1 (Supplementary Figs. 12a and 12b). POT1 Q623H has a slightly lower melting temperature, suggesting that this mutation is slightly less stable than WT POT1 (Supplementary Fig. 12b).

2. Experiments in Figs.4 and Figs.5 are problematic in terms of revealing mechanistic insight. The first part showed an amazing structure of POT1C with TPP1 PBM, and then the second part showed the impact of POT1C mutations on telomeres, but the mechanisms of action for these mutants are pretty much unknown. Are there mutations on POT1C that can affect the interaction with TPP1? Are there amino acids on POT1C, identified from the structure as Fig2 did, that are important for the interaction with TPP1 and can impact telomere function and genomic stability?

Based on the crystal structure, we now show that the POT1C residues W424, F438, Q508, D584, and Q623 are situated at the interface and directly interact with TPP1. Generating POT1 mutations W424E/F438R, designed to disrupt only the first binding module, completely abrogate the interaction between POT1 and TPP1 (Fig. 4h). In addition, engineered POT1 mutations Q580R/D584R/Q623H that interfere with both the second and the third binding modules also abolish POT1-TPP1 interaction (Fig. 4h). These mutations also completely abolish the telomere localization of POT1 (Supplementary Fig. 8).

In terms of the POT1C cancer mutations, we now show that POT1C residue Q623 directly interacts with TPP1, and that the Q623H mutation interferes with the interaction with TPP1 (Fig. 4h and Supplementary Fig. 8). This mutation cannot protect telomeres from activating an ATR-dependent DDR, resulting in inappropriate repair through the A-NHEJ repair pathway, generating genomically unstable chromosomal fusions that are tumor promoting (Figs. 5-7).

3. In addition to PBM, the authors used TPP1 RD mutants in their experiments. The authors

should better explain the RD domain. What is the relationship between RD and PBM? The RD domain appears just slightly larger than PBM.

RD represents the POT1 recruitment domain of TPP1 (TPP1 residues 244-337) and was first defined by the Songyang lab (Liu et al., *Nature Cell Biology*, 2004). In the revised manuscript we define the RD domain in the main text. TPP1^{ΔRD} has been well accepted as a dominant negative mutant of TPP1 in many previous studies (Liu et al., NCB 2004; Wu et al., Cell 2006; Guo et al., EMBO J 2007). As this reviewer pointed out, TPP1 RD is just slightly larger than the PBM of TPP1 (TPP1 residues 266-320) that is characterized by the yeast two-hybrid analysis in this study.

4. The authors utilized different techniques (SASX and XL-MS etc.) to study POT1-TPP1 interaction. It would be beneficial to the general audience to explain clearly the rationale for choosing these approaches as well as their pros and cons. The methods section needs to be improved.

Crosslinking mass spectrometry (XL-MS) is a valuable tool for providing information about protein folding and protein-protein interaction without high-resolution structures. Although XL-MS cannot compete with the level of detail and global information provided by traditional high-resolution methods like X-ray crystallography, NMR or Cryo-EM, it is more forgiving in term of sample concentration and purity. Additionally, XL-MS can be conducted *in vitro* under the condition that mimic native protein environment and capture interactions from dynamic states. XL-MS studies involve protein crosslinking through a chemical linker, digestion of the crosslinked protein complex into peptides, and identification of the crosslinked peptides, consequently, proximal residue pairs.

Small-angle X-ray scattering (SAXS) is a technique for low-resolution structural characterization of biological macromolecules in solution. SAXS can be used to probe proteins, nucleic acids, and their complexes without the need of crystallization and without the molecular weight limitations inherent in other high-resolution methods such as NMR or Cryo-EM. SAXS provides several key parameters of a biological macromolecule such as the molecular weight, radius of gyration R_g , maximum intramolecular distance D_{max} , and overall three-dimensional structure. Being complementary to the high-resolution methods, SAXS is often performed together with the high resolution techniques. These rationales for choosing XL-MS and SAXS in our study have been added in the method section of the revised manuscript.

5. The authors should also check their references to make sure the correct papers are referenced in the proper places. For instance, line 330, “localization of POT1 to telomere requires TPP1”, should reference original studies.

We have added the correct references in the revised manuscript.

6. In Fig 2g, TPP1 double mutant in the last lane has two bands, why? Fig 2g lacks a negative control. In Fig 2h, the input of lane 3 is too low, for both Myc-TPP1 and Flag-POT1, as was the Flag-POT1 input for lane 2. It made it difficult to interpret the IP data. The co-IP experiment should be repeated with proper controls and better data.

We have redone the co-IP experiments and the new data are now shown in revised Figures 4g and 4h.

7. Why was POT1 A364E used in Fig. 5b, but POT1 A364K used in the later experiments?

We thank the reviewer for pointing out this typo. It should be POT1 A364E and is corrected in the revised manuscript.

In Fig. 5e, input is low for both POT1 A364K and P371T compared to others. It is therefore difficult to conclude that the binding was at reduced levels.

Since our new cycloheximide chase data reveal that POT1 C terminal mutants A364E, P371T are extremely unstable even in the presence of TPP1 (Supplementary Fig. 9), it is not possible to load enough of these two proteins to achieve comparable input with the other mutants. However, despite this limitation we show that reduced levels of A364E and P371T mutants were still able to bind to ss telomeric DNA.

Why did Fig 6 not include mutants A364E and P371T? The data should be consistent and complete as in Fig 5.

We have now included analyses of mutants A364E and P371T in revised Figures 5, 6, and 7.

8. Line 140, the EMSA data of POT1HJRL with Holliday junction should be moved to supplemental figures.

The EMSA data of POT1 HJRL with Holliday junction is Supplemental Figure 2b in the revised manuscript.

Reviewer #3:

The authors demonstrate that mutant POT1 proteins have a dominant negative effect that impacts the ability of the POT1-TPP1 complex to prevent the telomere from eliciting a DNA damage response and DNA repair (Alt-NHEJ). The cancer link provides added interest to the structural study, although ultimately the actual mechanistic cause(s) of the resulting deficiencies in telomere maintenance remain unclear.

We now provide new data that provide mechanistic insights into why POT1C terminal mutations result in telomere dysfunction. Both A364E and P371T mutations result in POT1 misfolding, preventing interaction with TPP1 (Supplementary Fig. 12a). A cycloheximide chase assay reveal that these mutants exhibit reduced steady state levels and half-lives, even in the presence of TPP1 (Supplementary Fig. 9). POT1 A363E and P371T mutations therefore are unable to accumulate at telomeres, resulting in the activation of an ATR-dependent DDR and increased A-NHEJ chromosomal fusions that are potentially cancer promoting.

Introduction: The authors need to mention that the TPP1 OB fold is the domain that interacts with telomerase and they should provide a brief discussion of how TPP1 modulates telomerase activity as this information is needed later (see comments below about the discussion section).

We now add a paragraph to provide a brief discussion of how TPP1 modulate telomerase activity in the Introduction section.

The introduction needs some editing. Page 3, Lines 43-45: something is missing, perhaps the word, together.

We have edited this sentence as follows: In most eukaryotes, telomeres provide a solution to the end-replication problem, with telomerase, a specialized reverse transcriptase, adding telomeric repeats to the chromosome ends to ensure complete genome replication.

Lines 55-58: the difference between the functions of the two mouse POT1 proteins and the single human POT1 is not made clear.

We have edited this sentence as follows: There are two POT1 paralogs in mouse, mPOT1a and mPOT1b. mPOT1a functions primarily to repress an ATR-dependent DNA damage response at telomeres, while mPOT1b is required to repress nucleolytic processing of the 5' telomeric C-strand. The single human POT1 possesses both of these functions²⁵⁻³⁰.

Page 4 line 59, "protozoan" is missing, line 61, remove the capital T.

Corrected.

Figure 1a. The authors need to label the telomerase binding domain in TPP1 and the legend (or the figure) should make it clear that PBM stands for POT1 binding domain and TBP is TIN2 binding domain. Also, it would be helpful to indicate the residue numbers for the amino acids that delineate the various domains within TPP1 and POT1.

We label the TEL-patch (the telomerase interaction site of TPP1) of TPP1 in Figure 1a. The definition of PBM and TBM of TPP1 is now added in Figure 1 legend. We now add residue numbers in Figure 1a to define the boundaries of various domains in POT1 and TPP1.

Page 8, line 161: Mention that TPP1 residues 271, 275 and 279 are in helix H1.

We have made this correction. Now it reads "Four hydrophobic residues in helix H1 of TPP1_{PBM}, Leu271, Ala275, Leu279, and Leu281 make intimate interactions to the POT1 groove."

Page 10 line 209: change rooms to room. Line 212: add "the" to read the three strands....

We have made this correction.

Page 14 It would be helpful to explain the logic of using the F62A and Q623H mutations in the main text. Currently, this information is buried in the legend for Fig. 5.

We have clarified the logic of using these mutants in the revised manuscript.

Page 16 line 361 and Fig. 6 b-d. The terminology used to describe the frequency of chromosome fusions (Total fusion per chromosome or % fusions per chromosome) is most confusing and cannot be correct. Do the authors mean “number of fusions per 100 chromosomes? The authors must correct their terminology so that the frequency of fusions is correct and can be compared to fusion frequency reported for other mutants.

We thank this reviewer for raising this important point. The frequency of chromosome fusions is defined as the number of fusions divided by the number of chromosomes x 100%. We have put this into the MM section.

The experiment with SNM1b/Apollo^{-/-} MEFs (Fig 6d and all text relating to this figure) should be removed from the manuscript as it does not add useful information. The sudden switch to mouse mutants is unwarranted because (i) the mutants relevant to this manuscript (E572K and M587T don't have much effect on the fusion rate, (ii) it remains to be demonstrated that Apollo also processes leading strand telomeres in human cells and (iii) mouse Pot1 proteins are different from the single human POT1 protein so it is unclear how the Pot1a/b mutants relate to the human POT1 mutants.

We agree and have removed the SNM1b/Apollo^{-/-} MEF data from the revised manuscript.

Page 19 line 438: add the word protozoan to read “ciliated protozoan *O. nova*”.

Corrected.

Page 20, line 461: add the word is to read “and is protected by”

Corrected.

The discussion does not pack as much punch as it could. Additional discussion of how the POT1/TPP2pbd structure might impact telomerase action could strengthen it. Personally, I find it intriguing that the *O. nova* and human POT1/TPP1 complex contain so many of the same domains but they are re-arranged to deal with the different needs of the two organisms. The authors point out that human POT1/TPP1 needs to bind along the length of the 3' overhang. It would be helpful to point out that this not the case for *O. nova* where the overhangs are very short so they are likely bound by a single complex which is tailored to sequester the 3' end. It would also be helpful to mark the TEL-patch on TPP1 in Fig. 4G and to then point out how this appears to place the telomerase interaction site on a long arm that is distant from the POT1 ssDNA binding site. Perhaps this is what allows telomerase to access the DNA terminus and translocate as it adds repeats (see the Cech lab paper showing that a single POT1-TPP1 bound distant from the DNA 3' end can stimulate telomerase activity). I understand that the TIN2-binding domain of TPP1 is missing but it looks like this would sit closer to POT1 OB1 and OB2.

We thank this reviewer for raising these important points. We have revised the discussion section to include a short discussion of the difference between *O. nova* TEBP α - β and human POT1-TPP1 and pointed out that the overhangs in *O. nova* cells are very short so that they are likely bound by a single TEBP α - β complex to sequester the 3' end. In addition, we also provide a discussion about the potential functional significance of the extended V-shaped POT1-TPP1 complex in telomerase recruitment and activity regulation.

Reviewers' Comments:

Reviewer #1 (Remarks to the Author):

The revised manuscript is much improved and the authors have addressed all main points of criticism.

The figures are much improved as is the discussion that is now much more to the point.

Reviewer #2 (Remarks to the Author):

The authors have addressed my comments adequately.

Reviewer #3 (Remarks to the Author):

This revised manuscript is significantly easier to follow thanks the many small changes made in labeling and added explanations. The structural data have been strengthened in response to the reviewer's comments and this aspect of the manuscript is extremely strong. The revised discussion now highlights the true impact of the new structural information in terms of understanding how POT1-TPP1 may perform its various functions at telomeres.

Unfortunately, the enhanced clarity of the manuscript, and omission of some of the experiments with SNM1b/Apollo-/-MEFS, have revealed significant problems with the section of the manuscript dealing with the POT1 cancer mutations. The discovery of mutations in the newly identified C-terminal OB fold of POT1 is very interesting and important. But as described below, the in vivo analysis of these mutants lacks rigor and the results are heavily over interpreted. Consequently, this section of the manuscript does not provide much insight into the cancer promoting potential of these mutations.

Major Concerns:

The conclusions from the experiments performed with human cells rely on dominant negative effects caused by expression of mutant POT1 in the presence of endogenous POT1. However, the levels of expression of the mutant POT1 proteins are very different due to stability issues. As a result, it is unclear whether the lesser effects on telomere function observed with most unstable stable mutants merely reflects their lesser ability to compete with/displace the endogenous POT1. As an aside, the authors did not do a western blot with POT1 antibody to show the levels of expression of the various mutants relative to endogenous protein. This needs to be included. A usable commercial antibody to human POT1 is referenced in the recent de Lange lab paper

describing POT1 mutant that causes Coats Plus (Genes and development: 30, 1-15, 2016).

The authors attempt to circumvent the above problems by reconstituting mouse POT1a/b knockout cells with the human POT1 mutants. However, these experiments are problematic because mouse and human POT1 proteins are simply different so it is not reasonable to expect the in vivo effects of the mouse reconstitution experiment to exactly mirror what would be seen in human cells. Indeed, the human POT1 proteins do not rescue the mouse knockout in a manner analogous to the overexpression of WT or mutant human proteins in human cells. The discrepancy in the results with human and mouse cells could reflect differences between mouse and human shelterin/POT1, the low level of expression of the human mutants in human cells, or the complications of a dominant negative experiment. Either way, it is not valid to use the mouse experiments to draw conclusions about the “structural implication of POT1C mutations in human cancers”.

This reviewer feels very strongly that experiments with mouse POT1a/b reconstituted with human POT1 mutants do not contribute to the manuscript and should be removed because the results confuse the issue. Instead the authors should perform the equivalent experiment with a human POT1 knockout. The necessary conditional human POT1 knockout cells are available from the de Lange lab.

Although the authors have further examined the stability of the human POT1C mutants, it is still unclear how the C-terminal mutations would affect POT1 function to (maybe) promote cancer. The authors have clearly shown that some of the mutants decrease POT1 stability and hence reduce POT1 levels but others do not. They conclude that the low levels of the unstable proteins would lead to cancer by promoting genomic instability due to the inability of these proteins to protect telomeres from activating a DDR or aberrant repair, resulting in generation of chromosomal fusions (page 20). However, this conclusion is problematic on many levels.

(i) What about the stable POT1 mutants? How would they promote cancer? (ii) There is no evidence that overexpression of the least stable mutants leads to an increase in damage signaling in human cells. (iii) The POT1 mutation identified by the de Lange lab as the cause of Coats plus lies in POT1OB3. This mutation likely disrupts the interaction with the CST complex. Moreover, this mutation causes sudden telomere truncation. Thus, the fusions seen in current manuscript after overexpression of the (more stable) POT1 mutants could reflect a similar telomere truncation phenotype to that observed in the Coats plus mutation, rather than loss of telomere protection. (iv) The overall level of fusions observed in the human cells is very low and might or might not cause significant genomic instability.

The authors need to re-assess their in vivo data in light of the above comments and the de Lange lab Genes and Development paper. They then need to completely revise (or delete) the section on

“structural implications of POT1C mutations in human cancers” to remove references to the mouse experiments and capture the above points. The last paragraph of the discussion also needs to be heavily revised or deleted.

Other comments

Fig 5d Y axis and page 16 line 377 should read % of cells with reduced HA-POT1 at telomeres (i.e. not % of telomeres with POT1). According to the figure legend for Fig 5e, the authors counted the number of cells in which ≥ 5 HA-POT1 foci co-localized with telomeres. This is very different from counting the number of telomeres which displayed POT1 staining.

Point-to-point Responses to Reviewers' Comments

Reviewer #1:

The revised manuscript is much improved and the authors have addressed all main points of criticism.

The figures are much improved as is the discussion that is now much more to the point.

Thanks!

Reviewer #2:

The authors have addressed my comments adequately.

Thanks!

Reviewer #3:

This revised manuscript is significantly easier to follow thanks the many small changes made in labeling and added explanations. The structural data have been strengthened in response to the reviewer's comments and this aspect of the manuscript is extremely strong. The revised discussion now highlights the true impact of the new structural information in terms of understanding how POT1-TPP1 may perform its various functions at telomeres.

Unfortunately, the enhanced clarity of the manuscript, and omission of some of the experiments with SNM1b/Apollo-/-MEFS, have revealed significant problems with the section of the manuscript dealing with the POT1 cancer mutations. The discovery of mutations in the newly identified C-terminal OB fold of POT1 is very interesting and important. But as described below, the in vivo analysis of these mutants lacks rigor and the results are heavily over interpreted. Consequently, this section of the manuscript does not provide much insight into the cancer promoting potential of these mutations.

Major Concerns:

The conclusions from the experiments performed with human cells rely on dominant negative effects caused by expression of mutant POT1 in the presence of endogenous POT1. However, the levels of expression of the mutant POT1 proteins are very different due to stability issues. As a result, it is unclear whether the lesser effects on telomere function observed with most unstable stable mutants merely reflects their lesser ability to compete with/displace the endogenous POT1. As an aside, the authors did not do a western blot with POT1 antibody to show the levels of expression of the various mutants relative to endogenous protein. This needs to be included. A usable commercial antibody to human POT1 is referenced in the recent de Lange lab paper describing POT1 mutant that causes Coats Plus (Genes and development: 30, 1-15, 2016).

We see no compelling reason to examine the level of endogenous POT1, since previous reports by the deLange lab have shown that expression of mutant POT1 diminishes endogenous POT1 expression to almost undetectable levels (Loayza and de Lange, 2003). Instead, a better comparison is to express WT HA-POT1 in human cell lines and compare its properties to those of mutant HA-POT1 constructs, which is what we have done in Figures 5 and 6.

The authors attempt to circumvent the above problems by reconstituting mouse POT1a/b knockout cells with the human POT1 mutants. However, these experiments are problematic because mouse and human POT1 proteins are simply different so it is not reasonable to expect the in vivo effects of the mouse reconstitution experiment to exactly mirror what would be seen in human cells. Indeed, the human POT1 proteins do not rescue the mouse knockout in a manner analogous to the overexpression of WT or mutant human proteins in human cells. The discrepancy in the results with human and mouse cells could reflect differences between mouse and human shelterin/POT1, the low level of expression of the human mutants in human cells, or the complications of a dominant negative experiment. Either way, it is not valid to use the mouse experiments to draw conclusions about the “structural implication of POT1C mutations in human cancers”.

While we appreciate the reviewer’s concerns regarding the use of heterologous mouse/human systems to analyze POT1 functions, the de Lange lab has previously shown that human POT1 can be readily reconstituted and characterized in mouse cells lacking both POT1a and POT1b (Hockemeyer et al., 2007; Palm et al., 2009). Using POT1a/POT1b double knockout mouse embryo fibroblasts (MEFs), the de Lange lab showed that human POT1 possesses functional domains found in both POT1a and POT1b. We used a similar approach to characterize the functions of C-terminal mutant human POT1 in POT1a/POT1b double null MEFs. While WT human POT1 repressed TIF formation and end-to-end chromosome fusions, these cytogenetic abnormalities were common in cells reconstituted with C-terminal POT1 mutants (Figures 6c, 6d, 7d, and new Supplementary Fig. 11a).

To further address reviewer 3’s concerns, in the revised manuscript we also generated analogous mutations into mPOT1a and reconstituted WT or mPOT1a mutants into CAG-CreER; mPOT1aF/F, mPOT1b^{-/-} MEFs and removed endogenous mPOT1a with 4-HT. Like their human counterparts, expression of both mutants mPOT1a^{370E} and mPOT1a^{P377T} was reduced even in the presence of mTPP1 and these mutants cannot repress A-NHEJ mediated repair (new Supplementary Figs. 10b and 11b-c). Thus, the chromosome fusion data of mouse POT1a mutants is nearly identical to the data we observed when human POT1 mutants were reconstituted into these cell lines. These results indicate that there is no human-mouse species difference in terms of POT1 expression to complicate the interpretation of our data.

This reviewer feels very strongly that experiments with mouse POT1a/b reconstituted with human POT1 mutants do not contribute to the manuscript and should be removed because the results confuse the issue. Instead the authors should perform the equivalent experiment with a human POT1 knockout. The necessary conditional human POT1 knockout cells are available from the de Lange lab.

As stated above, the de Lange lab has on multiple occasions reconstituted human POT1 into MEFs lacking both POT1a and POT1b to decipher human POT1 function (Hockemeyer et al., 2007; Palm et al., 2009). This is a robust and well characterized method to study human POT1 in mouse cells devoid of endogenous POT1 proteins. The data is clear and we do not feel that the results confuse the issue.

Although the authors have further examined the stability of the human POT1C mutants, it is still unclear how the C-terminal mutations would affect POT1 function to (maybe) promote cancer. The authors have clearly shown that some of the mutants decrease POT1 stability and hence reduce POT1 levels but others do not. They conclude that the low levels of the unstable proteins would lead to cancer by promoting genomic instability due to the inability of these proteins to protect telomeres from activating a DDR or aberrant repair, resulting in generation of chromosomal fusions (page 20). However, this conclusion is problematic on many levels.

(i) What about the stable POT1 mutants? How would they promote cancer?

In Figures 7b, 7c, and 7d, we show that the stable POT1C mutants promote end-to-end chromosome fusions that are repaired through the mutagenic A-NHEJ pathway. Chromosome fusion induced genome instability is a hallmark of many human cancers.

(ii) There is no evidence that overexpression of the least stable mutants leads to an increase in damage signaling in human cells.

It is important to note that even the least stable POT1C mutations induce chromosomal instability, characterized as increased end-to-end fusions (Figure 7b, 7c) and chromatid fusions (Figure 7d).

(iii) The POT1 mutation identified by the de Lange lab as the cause of Coats plus lies in POT1OB3. This mutation likely disrupts the interaction with the CST complex. Moreover, this mutation causes sudden telomere truncation. Thus, the fusions seen in current manuscript after overexpression of the (more stable) POT1 mutants could reflect a similar telomere truncation phenotype to that observed in the Coats plus mutation, rather than loss of telomere protection.

The POT1 S322 mutation identified by the de Lange lab clearly affects telomere length maintenance, since expression of this mutant induced an increase in telomere length as well as an increase in the G-overhang (Takai et al., 2016). Although the construct for the POT1 C-terminal domain of POT1 (including both OB3 and HJRL motifs) contains POT1 residue 320 to 634, the first 21 residues (320-340) are disordered in the crystal structure. Therefore, residue S322 is actually in the linker region between POT1 N- and C-terminal domains.

Our POT1C mutations do not impact telomere length and do not induce telomere truncations, since telomeres are robust in cells expressing these mutants (Figure 7a). Telomere truncations would result in increased telomere-signal-free chromosome ends, which we never observe.

(iv) The overall level of fusions observed in the human cells is very low and might or might not cause significant genomic instability.

Please note that the number of telomere fusions scored is the total number of fusions divided by total number of chromosomes observed. While this yielded a relatively low number per chromosome (3-4%), we typically observe 1-2 fusions per metaphase in POT1C mutants (Figures 7a-7c). On a per metaphase level, this number of fusion is higher than we see in telomerase knockout mice (Artandi et al., 2000).

The authors need to re-assess their in vivo data in light of the above comments and the de Lange lab Genes and Development paper. They then need to completely revise (or delete) the section on “structural implications of POT1C mutations in human cancers” to remove references to the mouse experiments and capture the above points. The last paragraph of the discussion also needs to be heavily revised or deleted.

Given what we have presented, we believe that our data is valid and that our interpretation of the POT1C mutation data is correct. We feel that the mutational data is an important component of this paper, and adds important relevance to the structural studies.

Other comments

Fig 5d Y axis and page 16 line 377 should read % of cells with reduced HA-POT1 at telomeres (i.e. not % of telomeres with POT1). According to the figure legend for Fig 5e, the authors counted the number of cells in which ≥ 5 HA-POT1 foci co-localized with telomeres. This is very different from counting the number of telomeres which displayed POT1 staining.

Corrected.

References:

- Loayza D, De Lange T. POT1 as a terminal transducer of TRF1 telomere length control. *Nature*. 2003 Jun 26;423(6943):1013-8.
- Hockemeyer D, Palm W, Else T, Daniels JP, Takai KK, Ye JZ, Keegan CE, de Lange T, Hammer GD. Telomere protection by mammalian Pot1 requires interaction with Tpp1. *Nat Struct Mol Biol*. 2007 Aug;14(8):754-61. Erratum in: *Nat Struct Mol Biol*. 2009 May;16(5):572.
- Palm W, Hockemeyer D, Kibe T, de Lange T. Functional dissection of human and mouse POT1 proteins. *Mol Cell Biol*. 2009 Jan;29(2):471-82. doi: 10.1128/MCB.01352-08.
- Takai H, Jenkinson E, Kabir S, Babul-Hirji R, Najm-Tehrani N, Chitayat DA, Crow YJ, de Lange T. A POT1 mutation implicates defective telomere end fill-in and telomere truncations in Coats plus. *Genes Dev*. 2016 Apr 1;30(7):812-26. doi: 10.1101/gad.276873.115.
- Artandi SE, Chang S, Lee SL, Alson S, Gottlieb GJ, Chin L, DePinho RA. Telomere dysfunction promotes non-reciprocal translocations and epithelial cancers in mice. *Nature*. 2000 Aug 10;406(6796):641-5.